# Heterogeneity in *M. tuberculosis* β-lactamase inhibition by Sulbactam

Tek Narsingh Malla [1], Kara Zielinski [2], Luis Aldama[3], Sasa Bajt[4,5], Denisse Feliz[3], Brendon Hayes[6], Mark Hunter [6], Christopher Kupitz[6], Stella Lisova[6], Juraj Knoska[5], Jose Manuel Martin-Garcia [7], Valerio Mariani[6], Suraj Pandey [1], Ishwor Poudyal[1], Raymond G. Sierra [6], Alexandra Tolstikova[8], Oleksandr Yefanov[5], Chung Hong Yoon [6], Abbas Ourmazd [1], Petra Fromme [9], Peter Schwander [1], Anton Barty[8,10], Henry N. Chapman [4,5,11], Emina A. Stojkovic [3], Alexander Batyuk [6], Sébastien Boutet [6], George N. Phillips Jr. [12,13], Lois Pollack [2] & Marius Schmidt [1] ✉

For decades, researchers have elucidated essential enzymatic functions on the atomic length scale by tracing atomic positions in real-time. Our work builds on possibilities unleashed by mix-and-inject serial crystallography (MISC) at X-ray free electron laser facilities. In this approach, enzymatic reactions are triggered by mixing substrate or ligand solutions with enzyme microcrystals. Here, we report in atomic detail (between 2.2 and 2.7 Å resolution) by room-temperature, time-resolved crystallography with millisecond time-resolution (with timepoints between 3 ms and 700 ms) how the *Mycobacterium tuberculosis* enzyme BlaC is inhibited by sulbactam (SUB). Our results reveal ligand binding heterogeneity, ligand gating, cooperativity, induced fit, and conformational selection all from the same set of MISC data, detailing how SUB approaches the catalytic clefts and binds to the enzyme noncovalently before reacting to a *trans*-enamine. This was made possible in part by the application of singular value decomposition to the MISC data using a program that remains functional even if unit cell parameters change up to 3 Å during the reaction.

Beta(β)-lactamases are bacterial enzymes that provide multi-drug resistance to β-lactam antibiotics. They inactivate the β-lactam antibiotics by hydrolyzing the amide bond of the β-lactam ring[1,2] (Fig. 1). Our study focuses on β-lactamase from *Mycobacterium tuberculosis* (*Mtb*), the causative agent of tuberculosis. *Mtb* β-lactamase (BlaC) is a broad-spectrum Ambler class A[3] β-lactamase capable of hydrolyzing all classes of β-lactam antibiotics used for the treatment of tuberculosis[4,5]. In the latest report, the World Health Organization warned that

[1]Physics Department, University of Wisconsin-Milwaukee, Milwaukee, WI, USA. [2]School of Applied and Engineering Physics, Cornell University, Ithaca, NY, USA. [3]Department of Biology, Northeastern Illinois University, Chicago, IL, USA. [4]The Hamburg Centre for Ultrafast Imaging, Hamburg, Germany. [5]Center for Free-Electron Laser Science CFEL, Deutsches Elektronen Synchrotron, Hamburg, Germany. [6]Linac Coherent Light Source LCLS, SLAC National Accelerator Laboratory, Menlo Park, CA, USA. [7]Department of Crystallography and Structural Biology, Institute of Physical Chemistry Blas Cabrera, Spanish National Research Council (CSIC), Madrid, Spain. [8]Deutsches Elektronen-Synchrotron DESY, Hamburg, Germany. [9]School of Molecular Sciences and Biodesign Center for Applied Structural Discovery, 20 Arizona State University, Tempe, AZ, USA. [10]Center for Data and Computing in Natural Science CDCS, Deutsches Elektronen-Synchrotron DESY, Hamburg, Germany. [11]Department of Physics, Universität Hamburg, Hamburg, Germany. [12]Department of BioSciences, Rice University, Houston, TX, USA. [13]Department of Chemistry, Rice University, Houston, TX, USA. ✉e-mail: smarius@uwm.edu

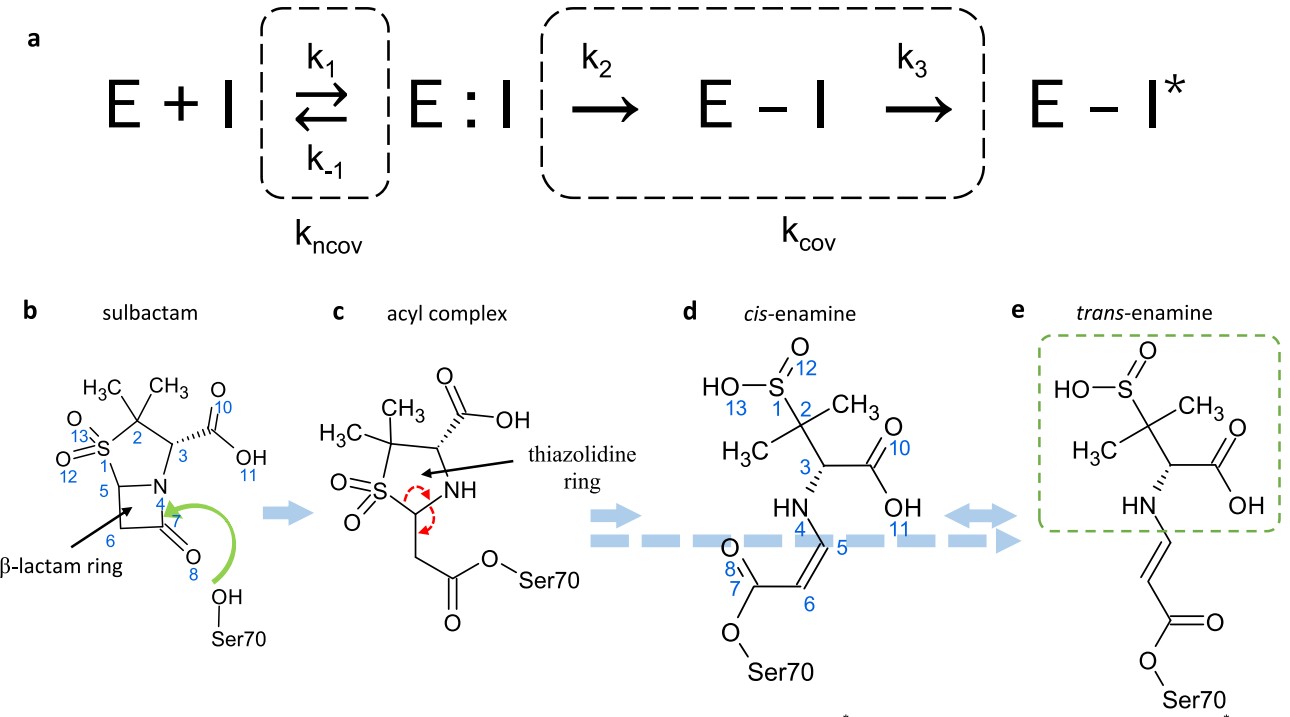

**Fig. 1 | A simplified two-step mechanism of BlaC inhibition by sulbactam. a** The first step is the formation of a noncovalent enzyme inhibitor complex (E:I) whose rate of formation depends on the concentration of the inhibitor in the unit cell and the rate coefficient ($k_{ncov}$). The reaction proceeds through a covalently bound (short-lived) acyl intermediate (E-I) and results in a product E-I*. **b–e** Structural view of the reaction. **b** The characteristic β-lactam ring is marked and the nucleophilic attack by Ser70 is shown by a green arrow. **c** The nucleophilic attack by the active serine opens the lactam ring of SUB leading to the formation of acyl-enzyme intermediate (E-I). In this current state, the E-I is very unstable and causes reorganization of bonds shown by dotted arrows. The E-I intermediate does not accumulate to become observable. The next step is the irreversible inhibition of enzyme by the chemically modified inhibitor (E-I*) which depends on the apparent rate coefficient ($k_{cov}$). The modification is the permanent opening of the 5-member thiazolidine ring and formation of either **d** cis-enamine (cis E-I*) and then to **e** trans-enamine (trans E-I*) or directly to trans-enamine following the blue dotted arrow. Cis- and trans-enamines differ in the configuration of the C5 = C6 double bond. A second nucleophilic attack by Ser128 on C5 may lead to cleavage of the fragment shown in the dotted box in **e**. Further modifications are possible which are not shown here.

multidrug-resistant tuberculosis is a public health crisis and a health security threat[6]. In 2020, tuberculosis was the 13th leading cause of death and the second leading infectious killer after COVID-19 (above HIV/AIDS), claiming 1.5 million lives worldwide[7]. The rapid and worldwide emergence of antibiotic-resistant bacteria, including *Mtb*, endangers the efficacy of antibiotics, causing the Centers for Disease Control (CDC) to classify several bacteria as urgent and serious threats. Research efforts focused on deciphering the molecular mechanism of antibiotic resistance within microbial pathogens such as *Mtb* can aid in drug design and resistance to β-lactamase action and therefore contribute to managing this evolving crisis.

Structural information involving enzyme-substrate reaction intermediates is essential for understanding the mechanism of action of enzymes. Although several static X-ray structures of BlaC with various substrates have been determined[4,8–13], the reaction intermediates of the β-lactamase reaction involving different antibiotic substrates remain mostly unknown. With time-resolved crystallography (TRX), conformational changes in biological macromolecules can be explored in real time[14]. Once a reaction is initiated within the crystals, ensemble-averaged structures are obtained along a reaction pathway. In addition, the chemical kinetics that govern the biological reaction can be deduced[15–19]. With mix-and-inject serial crystallography (MISC)[20], single-turnover enzymatic reactions can be investigated in a time-resolved manner. The substrate solution is mixed with enzyme microcrystals before the mixture is injected into the X-ray beam. The reaction is triggered by diffusion of the substrate[21–24], and the resulting change is probed after a delay by short X-ray pulses[25–28]. Using MISC, intermediates within the BlaC reaction with the third-generation

cephalosporin-based antibiotic ceftriaxone (CEF) were previously characterized from 5 ms to 2 s[20,25,29].

Irreversible enzyme inhibitors represent potential new drugs in the fight against antibiotic resistance. Sulbactam (SUB), clavulanate and tazobactam, all β-lactam based suicide inhibitors, irreversibly bind to β-lactamases and block their activity[30,31]. As a result, β-lactam antibiotics are protected from enzymatic degradation, which helps retain their efficacy. Due to its excellent solubility, SUB is a superb candidate for MISC experiments involving BlaC. Raman microscopy and mass spectrometry suggest a simplified mechanism for SUB binding to BlaC[32,33], as shown in Fig. 1a. The first step is the formation of a reversible (noncovalent) enzyme inhibitor complex in the active site (E:I). Following this association step, nucleophilic attack by the catalytically active serine on the β-lactam ring of the inhibitor (Fig. 1b) leads to the formation of a short-lived, covalently bound acyl-enzyme intermediate (E-I). Further modifications of the chemical structure of the inhibitor lead to an inactivated enzyme (E-I*). The structures of intermediates shown in Fig. 1 are relevant to the observed results presented in this paper within the measured timescale. SUB has also been described as a substrate of class A β-lactamases and can be hydrolyzed, albeit at a much slower rate than β-lactam antibiotics[30,34].

Numerous studies have shown that the pH of the macrophage compartment where the *Mtb.* bacterium resides is rather acidic at ~pH 4.5[35,36]. At low pH, the BlaC forms a dimer as determined by dynamic light scattering[25]. However, the biological significance of the dimer still has to be established. BlaC can be crystallized in a monoclinic space group with four subunits A–D (a dimer of dimers) in the asymmetric unit (Fig. 2a)[20,37]. The BlaC structure displays a large cavity of 30 Å in

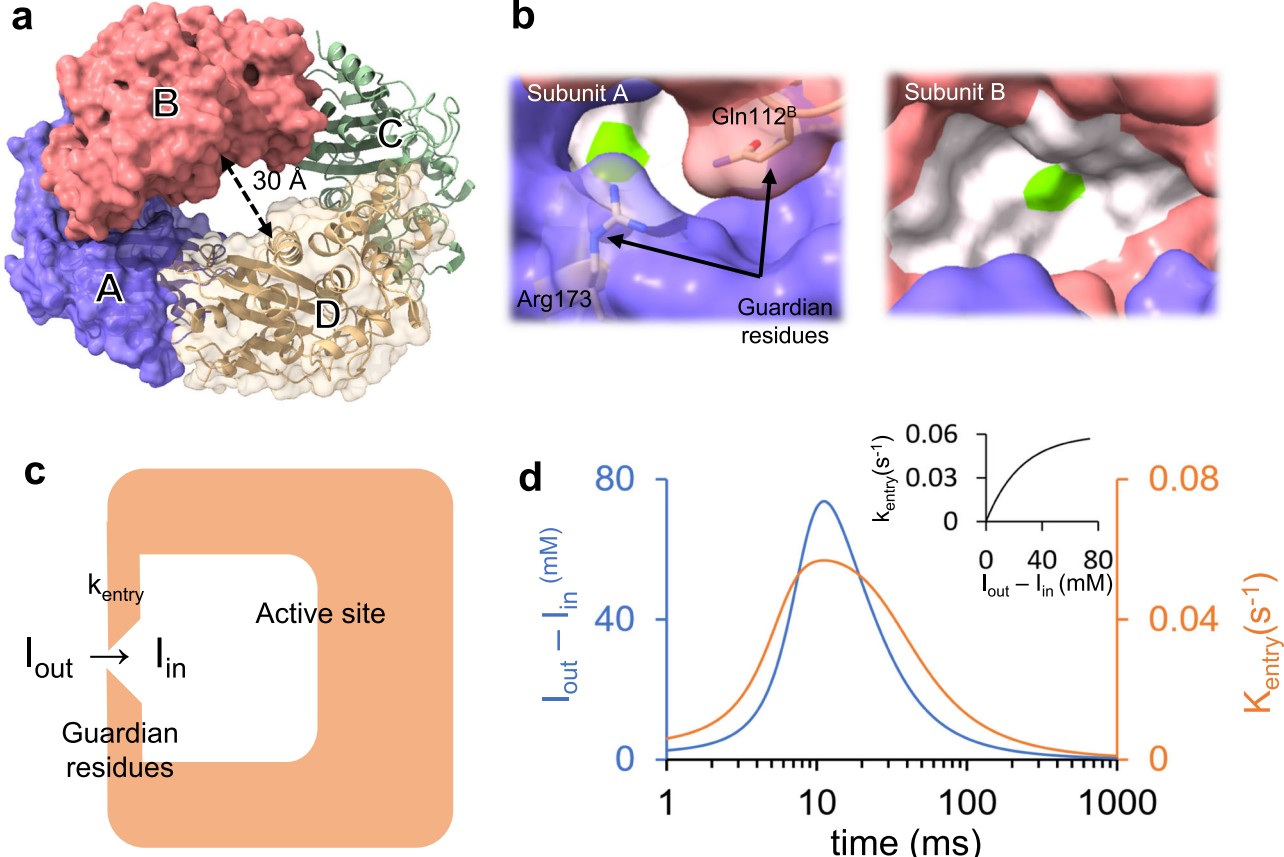

**Fig. 2 | Structure of BlaC and the gating mechanism. a** Subunits A – D in the asymmetric unit are marked and shown by blue, red, green, and yellow respectively. **b** Binding pockets of subunit A (left) and subunit B (right). The active site is represented by the white surface. The position of the catalytically active serine is marked in green. The access to the active site in subunit B is wide open. The entrance to the active site of subunit A is partially occluded by two residues (Gln112[B]

and Arg173) called the guardian residues. **c** Simplified scheme depicting the delayed entry of sulbactam into the active site through the guardian residues in subunits A/C. **d** Time-dependence of the concentration difference (blue line) and of the rate coefficient $k_{entry}$ (orange line). Inset: The dependence of $k_{entry}$ on the concentration difference.

diameter in the center. Additional cavities as large as 90 Å are identified within adjacent asymmetric units that allow easy diffusion of ligand molecules through the crystalline lattice (see ref. 25). The characterization of these channels[38] reveals that molecules as large as 15 Å are able to traverse the unit cells freely in all three directions. The E·I* structure of BlaC-SUB has been determined at cryogenic temperatures after soaking BlaC crystals for several minutes with SUB[13,39–41]. More recently, Pandey and colleagues captured an intermediate (presumably E:I) at a single time point (66 ms) after mixing BlaC crystals with SUB[29]. Although subunits B/D already displayed a covalently bound adduct, an intact, noncovalently bound SUB was observed in subunits A/C (Fig. 2a). Subunits A/C are rather inactive, since they did not participate at all in the reaction with CEF in earlier experiments[20,25]. Given the inactivity of subunits A/C, it was not clear whether the reaction with SUB takes more time to complete or proceed at all.

As in any time-resolved experiment, except in those performed on ultrafast time scales[42–45], multiple states can mix into any single time point observed during the reaction[46]. As demonstrated for X-ray data[16,] these mixtures can be characterized and potentially separated using singular value decomposition (SVD). Within this context, SVD is an unsupervised machine learning algorithm[47] that can inform the number of observable processes from time-resolved X-ray data, which is equivalent to the number of relaxation times and the number of structurally distinguishable time-independent reaction intermediates[16]. In addition, it can provide information regarding the kinetic mechanism and the energetics of the reaction[48–52]. SVD has

never been applied to a time series from an MISC experiment. The SVD method was developed to work with isomorphous difference maps assuming that the unit cells in the crystals essentially do not change during a reaction. An algorithm that takes changing unit cells into account is not available.

In this work, a time series of MISC data is analyzed to investigate both the binding of the SUB inhibitor to, and its subsequent reaction with, BlaC. However, the unit cell parameters of the BlaC crystals vary substantially after mixing (Supplementary Table 1). Therefore, a suite of programs, "pySVD4TX", is developed that remains functional even when the unit cell parameters change. With this we extract relevant times scales of the reaction kinetics which guide the subsequent structural interpretation of the time-resolved X-ray data.

## Results
### Binding of Sulbactam
The four subunits of BlaC found in the asymmetric unit are arranged in a shape reminiscent of a torus (Fig. 2a). The alternating subunits display similar binding kinetics while significantly differing from the adjacent subunits. Here, subunits A and C share similarities. As do subunits B and D. However, there are distinct differences between these pairs. Whereas the catalytic clefts of subunits B and D are wide open, those of subunits A and C are partially occluded by the neighboring subunits (Fig. 2b; Supplementary Movies 1 and 2). In particular, the residues Gln109[B/D] and Gln108[B/D] (the superscript B/D denotes residues from the neighboring subunits B and D, respectively) prevent substrate diffusion from the center of the torus, and Gln112[B/D] and

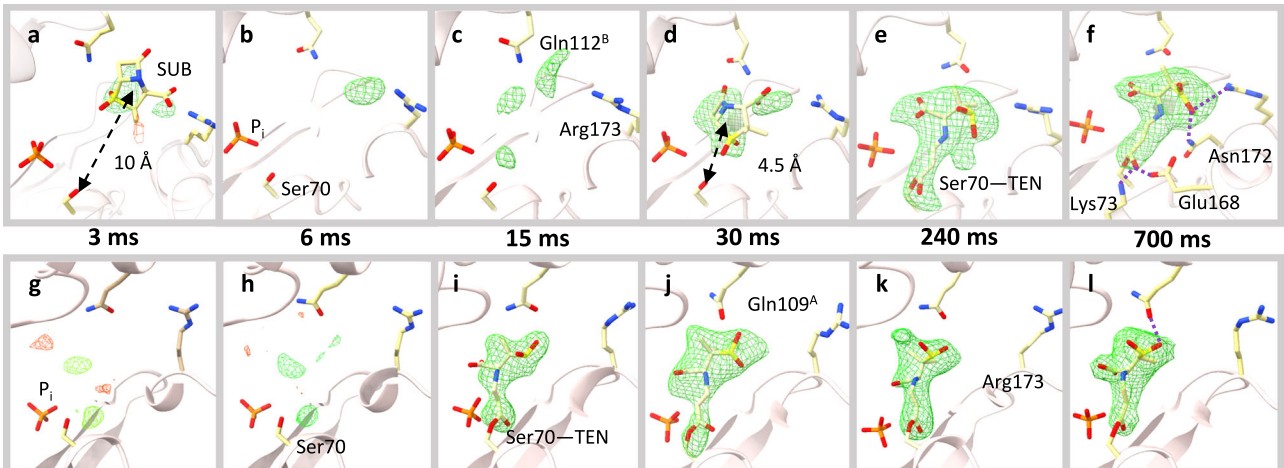

**Fig. 3 | Difference electron density (DED) maps in the active sites of subunit A and subunit B.** Omit maps are shown in all panels except for panels **d, f, j,** and **l** which show Polder maps (contour levels ±3σ). *Subunit A, top row:* **a** At 3 ms, weak densities can be identified at the entrance of cavity between Gln112[B] and Arg173 and a SUB placed there. (b) At 6 ms, very weak density is observed. The phosphate molecule (P$_i$) near the active site is marked. **c** At 15 ms, difference density features are identified closer to the catalytically active residue Ser70. The guardian residues (Gln112[B] and Arg173) that are located at the entrance to the binding pocket are marked. **d** At 30 ms a strong DED feature appears within the active site. An intact SUB molecule is placed there. **e** At 240 ms, the SUB has reacted with Ser70 to form TEN giving rise to an elongated density. **f** At 700 ms, the elongated density of the TEN is fully developed. Additional hydrogen bonds between the TEN and other side chains are shown. *Subunit B, bottom row:* **g, h** At 3 and 6 ms, no interpretable density was present in the catalytic center. **i** At 15 ms, the SUB has already reacted with Ser70 to from TEN. **j–l** TEN densities as observed at Δ$_{misc}$ from 30 ms to 700 ms. Gln109[A] and Arg173 are marked in **j** and **k**, respectively.

Arg173 block access to the active site from the exterior. In solution, the occlusion of the active sites of A and C most likely persists as the most plausible dimer arrangement is A-B (and C-D). This is justified by the number of interacting amino acid residues determined by ChimeraX[53]. About 30 residues are involved in the A-B and C-D dimer formation, whereas only 23 interact at the interface of the D-A and B-C dimers. The biological relevance of the tetramer is not clear. As in most crystallographic experiments, the observed interactions could be affected by the specific crystalline environment.

The reaction of SUB with BlaC was followed by difference electron density (DED) maps obtained at MISC delays (Δt$_{misc}$) of 3 ms, 6 ms, 15 ms, 30 ms, 270 ms, and 700 ms. As the datasets are non-isomorphous, simulated annealing mF$_{obs}$-DF$_{calc}$ omit maps and Polder maps were calculated as appropriate. At 3, 6, and 15 ms, the DED features near the active site of subunit A are weak (Fig. 3a–c). Substantial displacements of the residues flanking the active site are observed (Supplementary Table 2; Supplementary Movie 1). Particularly, the long-side chain residues such as Gln112[B] and Arg173 (called guardian residues, Fig. 2b) move outward from the active site before relaxing back to their original positions. Next to these residues at 3 ms, almost 10 Å away from Ser70, there are positive densities that are spatially more spread out than that of water. An SUB molecule can be placed in the electron density (Fig. 3a). However, refinement of the SUB is difficult (see Supplementary Fig. 1a for an explanation). At 6 and 15 ms, density features appear closer toward Ser70 (Fig. 3b, c). These results could indicate an initial trace of SUB molecules migrating to the active site after being held up by the guardian residues. Up to 15 ms, these densities are too weak that an SUB molecule can be placed with confidence.

At 30 ms, stronger DED features (max σ = 5.5) appear approximately 4.5 Å from the catalytically active Ser70 (Fig. 3d). An intact sulbactam can be modeled, which reproduces and corroborates the findings at 66 ms obtained from an earlier experiment[29]. The β-lactam ring is oriented away from Ser70. Between 66 ms and 240 ms, the SUB must rotate so that the β-lactam ring is positioned towards Ser70, at which point the nucleophilic attack occurs (Fig. 1b). At 240 ms, the elongated DED feature that originates from Ser70 directly supports the presence of a covalently bound *trans*-enamine (TEN) (Figs. 1e and 3e).

The BlaC-TEN adduct structurally relaxes until 700 ms, the final time point in the time series (Fig. 3f). These snapshots of the reaction in progress were assembled into a movie (Supplementary Movie 1) of an enzyme in action.

In subunit B, there is no evidence of an intact SUB that accumulates in the active site (Fig. 3g–h). Even the features that appeared near Arg173 in subunits A/C are not present. However, at 15 ms, the presence of a strong DED that extends from Ser70 supports a covalently bound TEN (Fig. 3i, Supplementary Fig. 1b). The occupancy refined to 85% in B and 86% in D. This suggests that the reaction is close to completion. At later time points, no large changes in the structure of the BlaC-TEN complex are apparent. (Fig. 3j–l). However, in subunits A/C, the reaction continues to progress, and structural changes in both the entire enzyme and the TEN are observed (Supplementary Movie 1).

## Temporal Variation of Difference Electron Density

SVD analysis is required to identify the number of intermediates as well as the relaxation times from the time series of DED maps[16] (see Methods). The right singular vectors (rSVs) obtained from the SVD analysis plotted as a function of MISC time delays represent the temporal variation of the reaction. It is important that the relaxation processes inherent to each rSV are accurately determined. The slow initial progress and sudden increase in the magnitude of DED values in the active sites require an appropriate function that could account for this behavior. An excellent fit was obtained by Eq. 2, which consists of a (step-like) logistic function that accounts for the steep first phase and an exponential saturation component. Detailed values of the parameters of Eq. 2 are shown in Supplementary Table 3.

The first two significant rSVs for each subunit are shown by blue and red squares, respectively (Fig. 4a–d). They allow for the determination of relaxation times τ₁ and τ₂ by fitting Eq. 2, shown by their respective colored solid lines. The rest of the rSVs, shown by colored diamonds in Fig. 4a, are distributed closely around zero and do not contribute to the subsequent analysis. The process with relaxation time τ₁ results from the first appearance of DED in the active sites. In subunits A/C, this process reflects the appearance of the noncovalently bound SUB, which occurs at approximately 23 ms and 25 ms, respectively. In subunits B/D, a covalently bound TEN is observed at 10 ms

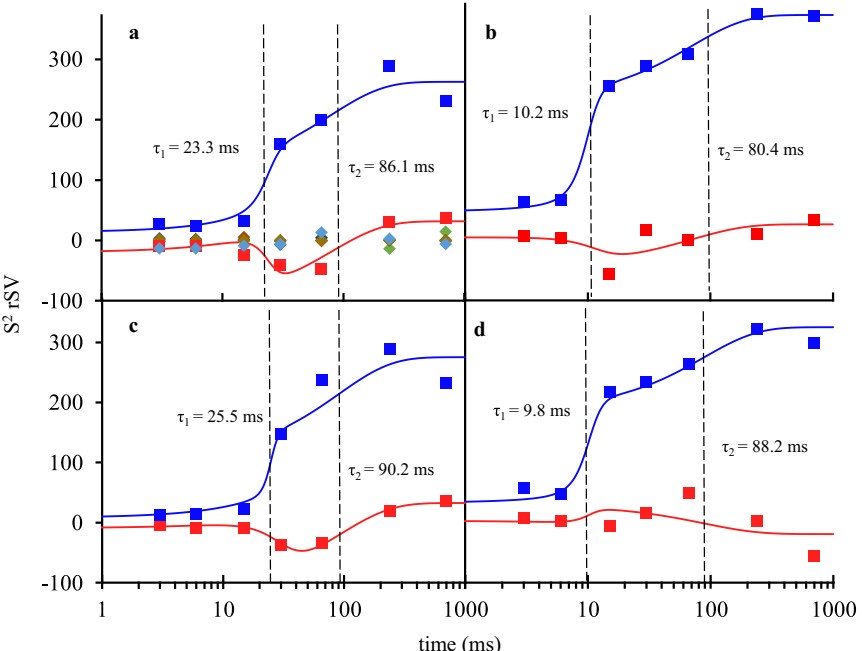

**Fig. 4 | Right singular vectors (rSVs) derived from a singular value decomposition of the time-dependent DED maps in the active sites of the BlaC. a** Right singular vectors plotted as a function of $\Delta t_{misc}$ for subunit A. The first and second significant rSV are shown by blue and red squares respectively. Solid colored lines are the result of a global fit of Eq. 2 to the significant rSVs. The colored diamonds represent insignificant rSVs. **b–d** Significant rSVs plotted as a function of $\Delta t_{misc}$ for subunits B, C and D respectively. Colors and lines as in **a**. The vertical dashed black lines in all panels denote the relaxation times $\tau_1$ and $\tau_2$ that result from the fit. For subunits A and C, $\tau_1$ belongs to accumulation of intact SUB in the active site, and $\tau_2$ corresponds to the formation of the covalently bound TEN. For subunits B and D, $\tau_1$ denotes the time when the reaction to TEN occurs and $\tau_2$ indicates a second relaxation phase. *Note*: The 66 ms data were obtained from a previous (published) experiment at the European XFEL where the experimental conditions were slightly different[29].

after mixing. In subunits A/C, the process $\tau_2$ results from the transformation of intact SUB to covalently bound TEN, which occurs approximately 85 ms and 90 ms after mixing. In contrast, in subunits B/D, TEN is already present during the first phase (Figs. 3g–l and 5a), and no further chemical modification of the TEN is observed. Despite this, a second relaxation process is also observed, which coincides with the SUB to TEN formation in subunits A/C.

## Inhibitor Diffusion

The kinetics of SUB binding to subunits B/D is evaluated first, since it allows for an estimate of the ligand concentration in the unit cell that is required to analyze the observations for subunits A/C. The total concentration of BlaC monomers in the crystals ($E_{free}$) is ~16 mM. If only subunit B is considered, the effective [$E_{free}$] is 4 mM. The apparent diffusion time of SUB into the microcrystals is ~7 ms (Table 1). It needs to be noted that MISC does not directly measure diffusion inside crystals. Instead, free ligand concentrations, and related to them the diffusion time, are estimated only indirectly through the rate equations (Eqs. 3 and 4), which ultimately must reproduce the observed (refined) ligand occupancies[29]. The unknown parameters in these equations must be varied until the concentrations of species [E:I] and [E-I*] best match their respective occupancy values (Fig. 5a). Additionally, relaxation times derived from the concentration profiles of intermediates must also agree with those obtained from SVD analysis of the DED maps (compare Table 1 and Supplementary Table 3). The concentrations of the free SUB within the microcrystals rise slightly slower than the values estimated in the central flow of the injector (Table 1, compare Fig. 5a blue solid line and blue squares). The use of a logistic function (Eq. 3) that describes the ligand increase in the unit cell is justified in particular by (i) the steep first phase observed in the rSVs (Fig. 4) but also by (ii) the very rapid increase of the ligand in the inner flow of the injector constriction where saturation occurs after 15 ms. The covalently bound TEN

accumulates rapidly with a characteristic time of 11.5 ms (Table 1), which is in line with the result from the SVD analysis (~10 ms). No additional intermediate is observed.

In subunits A/C, the apparent diffusion time of SUB necessary to reproduce the occupancies of the noncovalently bound intermediate is 20 ms (Fig. 5b and Table 1, $I_{in}$). This is much longer than that observed in subunits B/D (7 ms). This lag can only be explained by a restricted access to the active site. The guardian residues open the active site after approximately 6 ms (Supplementary Movie 2), which corresponds to the outside ligand concentration of approximately 35 mM (Fig. 5a). Entry to the active site is controlled by an additional rate coefficient, $k_{entry}$ (Fig. 2c). $k_{entry}$ was modeled (Eq. 5) by an exponential function that depends on the (time-dependent) concentration difference $\Delta I(t)$ between the outside and the inside of the active site and a characteristic concentration difference $\Delta I_c$ set to 25 mM (Table 1). $I_{out}$ is the SUB concentration in the unit cell, which is known from the substrate binding kinetics to subunits B/D (see above). After a delay of ~20 ms, SUB enters the active sites of subunits A/C, and $I_{in}$ rapidly increases. On the same time scale, the noncovalent E:I intermediate accumulates (compare $\tau_1$ in Figs. 4a, c and 5b). The noncovalently bound SUB triggers the next step of the reaction, which results in a covalently bound TEN. The concentration of TEN ([E-I*]) starts dominating [E:I] at 90 ms (Fig. 5b), which is in agreement with $\tau_2$ obtained from the SVD analysis (Fig. 4a, c). By 240 ms, more than 90% of the crystal is occupied by TEN (Fig. 5b).

While it is established that all the subunits take part in the reaction despite the heterogeneity, they do so at different rates. In subunits A/C, BlaC is inhibited by SUB via a two-step mechanism (Fig. 1a) which is observed in many other enzymes[54,55]. The noncovalent intermediate can accumulate since the SUB is not properly oriented (Fig. 3d) and can react to TEN only after an additional time delay. The Van der Waal's volume of SUB is 181.7 Å$^3$ with an average diameter of ~7 Å. The solvent accessible volume of the active site as determined by CASTp[56] is only

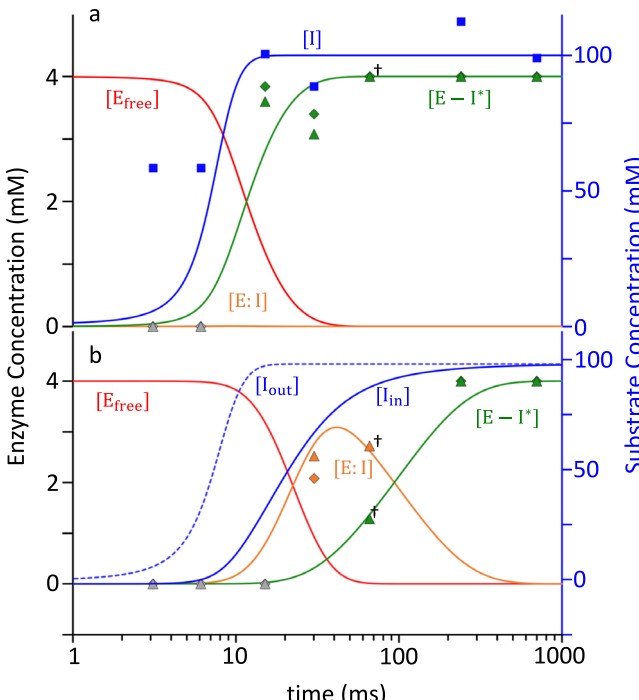

**Fig. 5 | Calculated concentration profiles of reactants and products in the active sites of BlaC compared to corresponding observables. a** Subunits B/D. Blue line: free SUB concentrations [I] in the unit cell. Blue squares: SUB concentrations in the central flow of the injector. Red line: time-dependent concentrations of the free BlaC [$E_{free}$]. Orange line: concentrations of the noncovalently bound SUB [E:I] intermediate (not observable). Green line: concentrations of the covalent enzyme-inhibitor complex TEN [E-I*]. Green triangles and diamonds: concentrations of E-I*, derived from refined ligand occupancy values in subunits B and D, respectively. Gray triangles and diamonds: SUB cannot be detected near the active sites. **b** Subunits A/C. Blue dotted line: free SUB concentrations [I] in the unit cell. Blue line: SUB concentrations [$I_{in}$] in the active site (note the delay relative to subunits B/D). Line colors as in **a**. Orange and green triangles and diamonds: concentrations of E:I and E-I* derived from refined ligand occupancy values in subunits A and B, respectively. †The 66 ms data point is included from a previous (published) experiment at the European XFEL[29].

**Table 1 | Characterization of SUB diffusion and reaction rate coefficients**

|  | Subunits A and C | Subunits B and D |
|---|---|---|
| **Diffusion parameters**[a] |  |  |
| $\mu$ | 0.59 ms$^{-1}$ | n.a. |
| $t_O$ | 7 ms | n.a. |
| $\Delta I_c$[b] | 25 mM | n.a. |
| $k_{max,entry}$[b] | 0.06 s$^{-1}$ | n.a. |
| **Rate coefficients**[c] |  |  |
| $k_{ncov}$ | 1.5 mM$^{-1}$s$^{-1}$ | 1.5 mM$^{-1}$ s$^{-1}$ (estimated) |
| $k_{cov}$ | 10 s$^{-1}$ | > 8000 s$^{-1}$ |
| **Characteristic times**[d] |  |  |
| $I_{out}$ | 7 ms (Fig. 5b blue dotted line) | 7 ms (Fig. 5a blue line) |
| $I_{in}$ | 20.1 ms (Fig. 5b blue line) | not observed |
| E:I formation | 21.2 ms (Fig. 5b orange line) | not observed |
| E-I* formation | 93.4 ms Fig. 5b green line | 11.5 ms Fig. 5a green line |

[a]Parameters to estimate the diffusion of SUB into the unit cells (Eq. 3).
[b]Parameters to estimate the flow of sulbactam into the occluded active site in subunits A/C as described by Eq. 5.
[c]Reaction rate coefficients of non-covalent and covalent intermediate formation.
[d]The approximate characteristic times are estimated from the half maximum of the curves shown in Fig. 5a, b.

~93 Å³ with widest gap between the surrounding residues being less than 10 Å. To react with Ser70, SUB must reorient to expose the β-lactam ring towards the active serine. SUB displacements will be restricted by interactions with the surrounding residues (Supplementary Table 2). The catalytic opening of the β-lactam ring and the unfurling of the thiazolidine ring all occur in succession inside the narrow reaction center cavity. This severely lowers the rate of TEN formation.

In subunits B/D, the covalent binding of SUB can apparently be explained by a one-step mechanism. However, the two-step mechanism described in Fig. 1a is also consistent with the observations when the noncovalent binding of substrate to the enzyme is much slower than the formation of TEN. By applying the two-step mechanism (Eqs. 1a and 4), a $k_{ncov}$ value of ~1.5 mM$^{-1}$ s$^{-1}$ and a large $k_{cov}$ value of ~8000 s$^{-1}$ are estimated (Table 1) so that the noncovalently bound E:I complex does not accumulate. The open binding pocket accommodates large chemical and structural changes that result in a large $k_{cov}$. The rate-determining step is controlled by $k_{ncov}$, although the noncovalent BlaC-SUB complex never accumulates. In a one-step scenario (the free SUB reacts directly to TEN), the pseudo one-step rate coefficient would be the same as the $k_{ncov}$. The two-step mechanism explains the enzymology in the active sites of all subunits in a consistent way. Efforts have been made to measure the kinetics of sulbactam reacting to other β-lactamases using initial rate measurement

and progress curve analysis methods[57]. These results showed that the rate coefficients for the inhibition of serine β-Lactamases varied widely. Even within class A β-lactamases such as the BlaC, the values differed by 4 orders of magnitude.

A second relaxation phase is observed in the SVD analysis of DED maps from subunits B/D (Fig. 4b, d), although the reaction to TEN has already taken place. It appears as if this is a result of the continuing reaction in subunits A/C and, related to this, an ongoing relaxation of the entire protein structure (Supplementary Movie 1). The relaxation times, $\tau_2$, of the second relaxation phase in all four subunits are all tied together between 80 ms and 90 ms (Supplementary Table 3), which is an indication for cooperative behavior of all subunits in BlaC. This observation would have been obscured without the prowess of SVD, which can track DED values across the entire reaction.

### Rapid reaction with Sulbactam in BlaC microcrystals

By taking into account the molecular volume that is enclosed by the Van der Waal's surface, SUB (181.7 Å³) is 2.5 times smaller than CEF (444.9 Å³). For CEF, 50% occupancy of the enzyme substrate complex has been observed at $\Delta t_{MISC}$ = 5 ms in subunits B and D[29]. Due to its smaller size, SUB should diffuse faster into the crystals than CEF, and the signature of an enzyme inhibitor complex is expected to appear at the earliest time points (3 ms and 6 ms). However, the first event that could be identified in the DED maps is the appearance of the TEN at 15 ms in subunits B/D (Fig. 3i). The concentrations of SUB at the active site of subunits B/D, which allow direct, unrestricted access, can be taken as an estimate of the SUB concentration inside the BlaC crystals. The slight lag between the estimated SUB concentration in the inner flow of the injector constriction and the concentrations in the crystals at early time points (Fig. 5a, compare blue squares with the blue line) is expected since diffusion in BlaC crystals is slowed down in crystals[29] compared to water. The apparent diffusion time (7 ms, Table 1) of SUB is almost the same as that for the antibiotic substrate CEF ( ~ 5 ms)[29]. By this time, the concentration of SUB is more than 3 times the concentration of BlaC molecules in the crystal. Nevertheless, contrary to expectations based on the size of the SUB, no electron density is present at the earliest time points. This can be explained by the smaller second-order binding coefficient, $k_{ncov}$ (1.5 mM$^{-1}$ s$^{-1}$ for SUB compared to 3.2 mM$^{-1}$ s$^{-1}$ for CEF), which prevents the accumulation of electron density at earlier times.

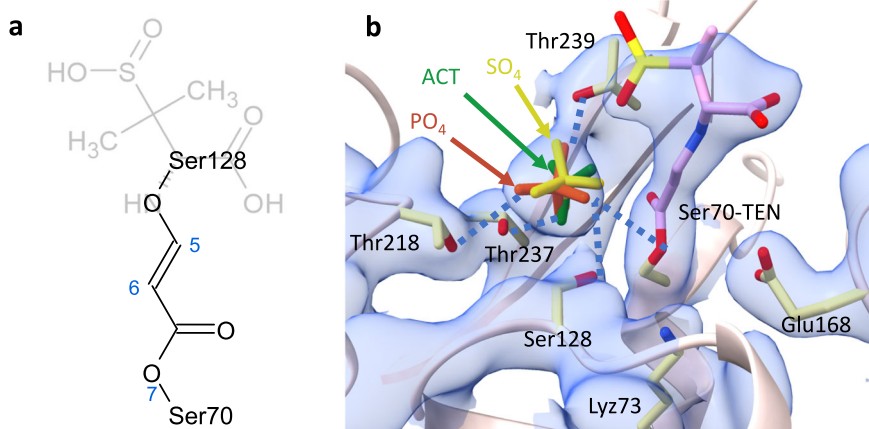

**Fig. 6 | *Trans*-enamine on longer timescales. a** Chemical structure of the TEN after the formation of cross-linked species. The leaving group is shown in pale color. **b** A 2F$_{obs}$-F$_{calc}$ map (blue, 1σ contour level) is shown near the active site of subunit B at Δt$_{misc}$ = 240 ms. Key active site residues and the TEN (purple) are marked. The sulphate (SO$_4$, yellow) and acetate (ACT, green) observed in other β-lactamases (PDB entries 5OYO and 7A71, respectively) are overlayed. Hydrogen bonds established by phosphate (PO$_4$, orange) with TEN and surrounding residues are depicted by blue dotted lines. TEN might not be able to get close to Ser128 without displacing the phosphate.

The characteristic times observed in both classes of subunits for the formation of the covalently bound TEN species (approximately 10 ms and 90 ms, respectively, Fig. 5) are quite fast in comparison to earlier suggestions that the reaction might take minutes to complete[13,39]. The fast reaction provides an advantage when β-lactam substrates and the SUB inhibitor are competing for the same active site. For example, the noncovalent enzyme-substrate complex with CEF persists for up to 500 ms[25]. During this time, CEF can leave the enzyme and be replaced first competitively and then irreversibly by a quickly reacting SUB molecule. Since the inhibitor competes with co-administered antibiotics for the active site of BlaC, one can imagine that the covalent bond formation with an inhibitor must occur as fast as possible to effectively eliminate β-lactamase activity in the presence of substrate. This is in addition corroborated by Jones and coworkers, who reported that SUB has a ten times higher affinity and binding constant for plasmid-mediated class A β-lactamases than cefoperazone[58], which is a third-generation cephalosporin-based antibiotic similar to CEF.

### Ligand gating, induced fit and conformational selection

Our results show that ligand binding to enzymes may be more complicated than initially thought[13,32,33]. Only after a delay does the ligand penetrate into the active sites of subunits A/C controlled by the guarding residues (Fig. 5b). The narrow entrance to the active site (Fig. 2b; Supplementary Movie 2) and the displacements of the guardian residues (Supplementary Table 2; Supplementary Movies 1 and 2) are reminiscent of a substrate tunneling-and-gating[59-61]-like mechanism that has not yet been discovered in published structures of BlaC. Supplementary Movie 2 shows that the movement of the guardian residues opens the entrance, thereby controlling ligand access. More work is needed to determine the mechanism that drives the displacement of these residues. As these displacements were not observed when reacting with CEF[25,29], an allosteric mechanism that links the position of the guardian residues to the covalent binding of SUB in adjacent subunits is unlikely. However, electrostatic interactions[62] of the negatively charged sulbactam with the positively charged Arg173 or even polar interactions with Gln112$^{B,D}$ may plausibly induce these structural changes.

At Δt$_{MISC}$ of 30 ms and 66 ms[29] SUB occupies the active sites of subunits A/C without reacting with Ser70. During this time, conformational changes of BlaC are apparent (Supplementary Movie 1) to accommodate the SUB. This resembles an induced fit[63]. Similar observation was made by Yi and colleagues[64]: With different mutations they demonstrated that the omega loop (residues 161–179) in β-lactamases relaxes to allow access of substrates and regroups upon substrate binding. The Arg173 guardian residue discussed here lies on the tip of the loop and is one of the closest residues to the catalytically active Ser70. The unfavorable SUB orientation prevents the direct attack of the active Ser70 towards the β-lactam ring. Ser70 can react with the β-lactam ring only after a rotation of the SUB. Fluctuations of Ser70 towards SUB carbon C$_7$ will then lead to a very short-lived, indiscernible "transition state-like" structure E-I with a covalent bond between SUB and BlaC. This is a different form of conformational selection[65,66] in the sense that there is not a particular "preferred" protein conformation that reacts with a substrate, but here, a particular (active) ligand orientation is required and "selected" by the enzyme for further reaction. The rate of reorientation (rotation of the SUB) seems to control the rate of this reaction. Once a favorable orientation is reached, further reaction to TEN is instantaneous on the timescale of the observation (90 ms, Table 1). This informs the design of improved (faster) inhibitors that consist of symmetric active moieties[67,68] or are engineered to enter the active sites in the correct orientation.

In subunits B/D, the inhibitor is brought in rapidly by diffusion and reacts instantaneously (<1.5 ms) on the timescale of observation (> 15 ms). Neither an induced fit nor conformational selection can be observed or distinguished, which previously led to intense discussions for other enzymes[69]. The active site structures relax in unison with the extent of covalently bound inhibitor in all subunits, as explained above.

### The fate of the *Trans*-Enamine

It has been proposed that on longer timescales (> 30 min), a second nucleophilic attack by a nearby serine can occur in other, structurally closely related Ambler Class A β-lactamases[30]. This serine (Ser128 in BlaC) may react with the C5 position of the TEN (Fig. 1). This is followed by the loss of the opened thiazolidine ring fragment (Fig. 1d). A covalent bond may be formed between C5 and Ser128 of BlaC (Fig. 6a), leading to prolonged inhibition of the enzyme[30,39]. It has also been suggested that only transient inhibition by TEN is responsible for the medical relevance of SUB, as any reaction that lasts longer than one hour is irrelevant due to the bacterial lifecycle of ~30 min[39]. However, the life cycle of *Mtb* is ~20 h[70]. The permanent inhibition achieved only after the second nucleophilic attack might be the ultimate factor for SUB's clinical usefulness in fighting antibiotic resistance in slow-growing bacteria such as *Mtb*. Inspection of the soaked structure may

provide an answer. Covalently bound TENs were observed in all four subunits when BlaC was soaked with SUB for 3 h (Supplementary Fig. 1c–f). The B-factors of the fragment beyond N4 that would be cleaved off (displayed in pale colors in Fig. 6a) are consistently higher by 20 Å$^2$ than that of the part that would form the cross-linked species. However, it is more plausible that higher B-factors are caused by the dynamic disorder of the long TEN tail and not by the presence of a mixture of intact and fragmented TEN. There is no clear evidence of TEN fragmentation, and TEN remains the physiologically important species for BlaC inhibition for hours.

A phosphate group binds to a specific site immediately adjacent to the active serins 70 and 128 (Fig. 6b) with multiple hydrogen bonds to surrounding residues. This location appears to be conserved among all published BlaC structures, whereas others have also reported sulfate and acetate molecules in the same position (Fig. 6b)[13,71–73]. Naturally occurring compounds with phosphate-like groups, such as adenosine phosphates, might also interact with BlaC. β-lactamase production increases in some bacteria grown in a phosphate-enriched medium[74], while phosphate can also promote the hydrolysis of the clavulanate inhibitor by BlaC[73]. More structures are required after soaking with high SUB concentrations for longer periods of time, perhaps days, to observe potentially fragmented TEN. Since the phosphate is replaceable[20,25,29], the TEN might indeed react further. Larger inhibitors such as tazobactam and clavulanic acid might also be able to displace the phosphate molecule directly. Recently, *trans*-enamine intermediates of tazobactam were identified in the serine β-lactamase TEM-171 at a position similar to that of TEN in BlaC and at another that is occupied by the phosphate in BlaC[75]. The diverse chemistry that is already observed very early on in BlaC may extend to much longer time scales.

## Discussions

There are other classes of β-lactamases that are more concerning than BlaC, such as metallo β-lactamases (MBL). They are capable of hydrolyzing almost all clinically available β-lactam antibiotics and inhibitors[76,77]. Similar work to the one presented here and earlier[25,29] could be performed on MBLs to gain more structural insight into their catalytic mechanisms. Time-resolved pump-probe crystallographic experiments using a caged Zn molecule[78] already show how the antibiotic moxalactam is inactivated by MBL on a timescale longer than 20 ms. To characterize the important substrate binding phase on single ms and even sub-ms time scales, it would be desirable to follow this or a similar reaction as well as the binding of a MBL inhibitor with MISC.

MISC is a straightforward way to structurally study enzyme function. Reactions are visualized in real time as snapshots of the enzyme in action. Our results provide insight into how the shape of the active site determines the rate coefficients and reaction mechanisms of a biomedically relevant reaction. From the MISC data, diffusion times and rate coefficients can be estimated by applying informed, chemically meaningful constraints and tying the analysis to observed occupancy variations of transient enzyme-inhibitor complexes in the different subunits. The key advantage of MISC is that it provides at the same time a localized view of processes that unfold in individual subunits and a global perspective of the behavior of the entire molecule with near atomic precision. High repetition rate XFELs[79,80] and upgraded synchrotron light sources[81,82] will facilitate the collection of time series that consist of very closely spaced MISC time delays. At synchrotrons where liquid injection is difficult to achieve, other techniques to expose microcrystals to the X-ray pulses may be employed. These include depositing crystals on fixed targets[26] or on tape drives[83,84]. A global evaluation of these time series assisted by a principal component analysis such as SVD and other machine learning techniques[45,47] will provide a detailed and direct view into enzyme catalysis and inhibition.

## Methods

### Sample preparation

BlaC was expressed in *E. coli* as previously described[25]. In short, *E. coli* cultures were grown in terrific broth at 37 °C to a OD600 value of 0.8. Overexpression was induced by 1 mM isopropyl-β-d-thiogalactopyranoside (IPTG) after lowering the temperature to 20 °C. After 3 hours, the culture was induced a second time with 1 mM IPTG, and shaken overnight at 20 °C. The mature cells were harvested by centrifugation. Cells were resuspended in lysis buffer (20 mM Tris Base, 150 mM NaCl, pH 8.). After lysis of the bacterial cells, debris was centrifuged at 50,000 g for 1 h. The lysate was then pumped through a column containing 15 mL of Nickel resin. The resin was washed with a buffer consisting of 20 mM Tris Base, 300 mM NaCl, 20 mM Imidazole, pH 8 (about 20 column volumes each). The final product was eluted with 300 mM imidazole and dialyzed immediately in 20 mM Tris base, 200 mM NaCl, pH 8. The protein was further purified through size exclusion chromatography using Superdex 200 resin.

Crystals were grown on site at the Linac Coherent Light Source (LCLS) as published[25]. In short, purified BlaC protein (concentrated to 150 mg/ml) was added dropwise to 2.4 M ammonium phosphate (AP) at pH 4.1 while stirring until a ratio of 1:9 (protein: precipitant) was reached. The stirring was stopped after ~12 h. Protein microcrystals were left to mature for 2 days at room temperature. As the crystals settled, the supernatant was removed to reach the desired concentration. To avoid potential clogging of the injector nozzles, the crystal slurry was filtered through a 20-micron syringe filter. The final concentration of crystals loaded into the injector reservoir was $5 \times 10^9$ crystals/ml. For mixing, a solution of 150 mM SUB in 50 mM AP at pH 4.5 was prepared. A reference (unmixed) dataset was obtained by mixing the BlaC microcrystals with water.

### Data collection and processing

Data were collected at the Macromolecular Femtosecond Crystallography (MFX) instrument[85,86] at the LCLS in October 2020 [beamtime lu6818]. The XFEL was operating at 120 Hz with a pulse energy of ~9.8 keV (λ = 1.26 Å) and a pulse duration of 20–40 fs. The XFEL beam was focused to a spot size (FWHM) of 3μm. The crystal slurry was mixed with SUB solution and injected into a helium filled chamber at ambient temperature. The injectors allow simultaneous mixing and serial injection of sample into the X-ray interception region. Injectors were provided by Pollack group (Cornell University) and are based on the design published by Calvey et al.[87,88]. A central thin stream of a suspension containing microcrystals flows down a delay line of variable length, called the constriction (Supplementary Fig. 3). A solution of substrate or ligand concentrically surrounds the central stream allowing for rapid diffusion into the microcrystals within the stream[88]. At the end of the constriction, the mixture is injected into the X-ray interaction region. The geometry of the injectors and the sample delivery parameters are shown in Supplementary Table 4.

Our BlaC microcrystals are quite small (~10 × 10 × 2 μm$^3$) and contain large solvent channels (Fig. 2a), both of which promote rapid diffusion of substrate into the crystals. The time delay (Δt) in MISC is the time taken by the crystals to travel from the mixing region in the injector to the X-ray interaction region. "Time point" and "time delay" are used interchangeably throughout this manuscript and denoted by Δt$_{MISC}$. SUB is provided at a concentration of 150 mM which is much higher compared to the 16 mM concentration of BlaC in crystals. Accordingly, stoichiometric concentration is reached on a time scale much faster than the diffusion time[24,29]. Different injector designs allowed data acquisition at 6 different Δt$_{MISC}$ from 3 ms to 700 ms (Supplementary Table 1).

Diffraction patterns (DPs) were collected with the epix10K2M detector[89] at 120 Hz. The experiment was monitored in real time with the OnDA (online data analysis) Monitor (OM)[90], which also provided feedback on hit rate and spatial resolution. The raw data were

processed by Cheetah[91]. Cheetah identifies DPs containing potential Bragg reflections (hits). In addition, it has a built-in masking tool that allows masking of unwanted detector pixels. The selected patterns were further processed by the CrystFEL suite of programs[92,93]. *Indexamajig* was used to index and integrate the DPs using a combination of *mosflm, dirax, xds, asdf* and *xgandalf*[94–97]. The detector geometry and distance were refined by *geoptimiser*[98]. Merging and scaling of the intensities were performed with *partialator* using the partiality model *xsphere*[99]. Figures of merit and other data statistics were calculated using *compare_hkl* and *check_hkl*.

To investigate SUB binding to BlaC on a longer time scale, macroscopic crystals were grown in sitting drops (10 μl BlaC at 45 mg/ml⁻¹ mixed (1:1) with 2.1 M AP at pH 4.1). Crystals were soaked for 3 h in a cryobuffer consisting of 2 M AP, 20% glycerol and 100 mM SUB. The crystals were flash frozen in liquid nitrogen and investigated at beamline ID-19 of the Structural Biology Center, Advanced Photon Source, Argonne National Laboratory. Data were collected by the proprietary *sbccollect* program[100] and processed by *HKL-3000*[101]. The data collection and refinement statistics are listed in Supplementary Table 1.

## Difference maps and structure determination

Because of the change in unit cell parameters, omit difference electron density maps were calculated throughout. First, a reference (unmixed) structure was refined using the structure published in the protein data base (PDB)[102] entry 6B5X as the initial model. For the analysis of the MISC data, the water and the phosphate molecules in the active sites were removed from the reference structure. For each $\Delta t_{MISC}$, the resulting structure was refined using the standard simulated annealing (SA) protocol in *Phenix*[103] against the experimentally observed structure factor amplitudes $|F_{obs}(t)|$. Following refinement, $m|F_{obs}(t)| - D|F_{calc}|$ omit maps ($DED_{omit}$) were calculated for each time point, where $|F_{calc}|$ are calculated from the refined model. Polder difference maps (PDMs)[104] were calculated to display weak ligand densities in some time points to assist in the placement of the ligand. To calculate a PDM, a suitable small molecule (which was SUB in our case) is placed at the center of the active site at a position where the ligand density is expected to be. The algorithm then generates an omit map by excluding bulk solvent modeling around the selected region. This way, weak densities become apparent, which may otherwise be obscured by the bulk solvent. PDMs were calculated particularly using the cryosoaked data (Supplementary Table 1).

Both the unbound sulbactam and the covalently bound *trans*-enamine (TEN) were manually modeled into the DED maps using *Coot*[105]. The ccp4-program *AceDRG*[106] was used to generate coordinates and restraint for the covalently bound TEN. Several cycles of refinement in *Refmac*[107] and *Phenix* followed by manual inspection and re-modeling with *Coot* led to models with excellent R-factors. Ligand occupancies were determined by 'group occupancy refinement' in *Phenix*. A value of 100% indicates that stoichiometric concentration was reached. The refinement statistics are listed in Supplementary Table 1. All figures that display DED maps in the main text and the supplementary material as well as the molecular movies (Supplementary Movies 1 and 2) were prepared using *UCSF ChimeraX*[53]. The chemical structure drawing software *ChemSketch* (ACD/Labs) was used to create chemical schematics.

## Singular value decomposition

A kinetic analysis was performed by the application of singular value decomposition (SVD) to the X-ray data[16] using $DED_{omit}$ maps (for data collection, data processing, difference map calculation and structure determination, see the Supplementary Information, SI). A region of interest (ROI) was determined that corresponds roughly to the volume of an individual active site selected from the four subunits in the asymmetric unit. The ROI covers the volume occupied by the SUB and TEN ligands and the entire side chain of Ser70. It touches the terminal

atoms of residues Lys 73, Glu168, Thr239, Asn172, Arg173, Ser128, Ser102 and Gln112 (from B/D) in the case of subunits A/C. Difference electron density values found in grid points (voxels) of the ROI were assigned to an m-dimensional vector. N = 6 of these vectors were obtained for measured $\Delta t_{misc}$ and assembled into an m x n dimensional matrix A, called the data matrix. SVD is the factorization of matrix A into three matrices: U, S, and $V^T$ according to

$$A = U \cdot S \cdot V^T \tag{1}$$

The columns of the m × n matrix U are called the left singular vectors (lSVs). They represent the basis (eigen) vectors of the original data in data matrix A. S is an n × n diagonal matrix, whose diagonal elements are called the singular values (SVs) of A. These nonnegative values indicate how important or significant the columns of U are. The columns of the n × n matrix V, called the right singular vectors (rSVs), contain the associated temporal variation of the singular vectors in U. S contains n singular values in descending order of magnitude.

The number of significant singular values and vectors can inform how many kinetic processes can be resolved[16,108]. The SVD results can then be interpreted by globally fitting suitable functions to the rSVs, which can consist, in the simplest form, of sums of exponentials. The rSVs contain information on the population dynamics of the species involved in the mechanism[16,108].

The earlier program SVD4TX[16] and a newer version[109] could only be applied to isomorphous X-ray datasets since these implementations relied on a region of interest that is spatially fixed. This is not given when the unit cell changes. When the change in unit cell parameters is greater than 1/4 of the maximum resolution, the datasets are considered non-isomorphous[110]. To accommodate changing unit cells, a new approach was coded by a combination of custom bash scripts and python programs described below.

## Adapting SVD for MISC datasets with changing unit cell parameters

The DED map is calculated (Supplementary Methods) in the CCP4 file format[111] and covers the entire unit cell of the crystal. The maps are represented by a three-dimensional (3D) array with $m_x$, $m_y$, and $m_z$ grid points for each unit cell axis. Each 3D grid point (voxel) contains the magnitude of the difference in electron density at that given position (Fig. 7a). In such a DED map, positive features indicate regions where atoms have shifted away from their position in the reference model. Negative features are then found on top of the atoms in the reference model. Most of the map contains only spurious noise except in ROIs such as the active sites where larger structural changes are expected due to the binding or dissociation of a ligand (Fig. 7b). The noise within the majority of the difference map would interfere with the SVD analysis. To avoid this, an ROI was isolated individually for each subunit, and an SVD was performed only on the DED within. When multiple active sites are present, each active site can be investigated separately. Supplementary Fig. 2 shows a flow chart of the steps required to prepare data matrix A. The steps are described in detail below.

Step 1: The coordinates of the atoms of the amino acid residues and the substrate of interest are specified in a particular subunit. This defines the ROI. For the present work, four different ROIs were defined, one for each subunit A to D, and investigated separately.

Step 2: A mask is calculated that covers the selected atoms plus a margin of choice (Fig. 7b). The density values outside of the mask are set to 0, while the ones inside are left unchanged. This results in a masked map with the dimensions of the original map with density values present only around the emerging DED in the active site (Fig. 7c). This mask evolved later (after step 4) by allowing only grid points that contain DED features greater or smaller than a certain sigma value (for example, plus or minus 3 σ) found at least in one time point[16].

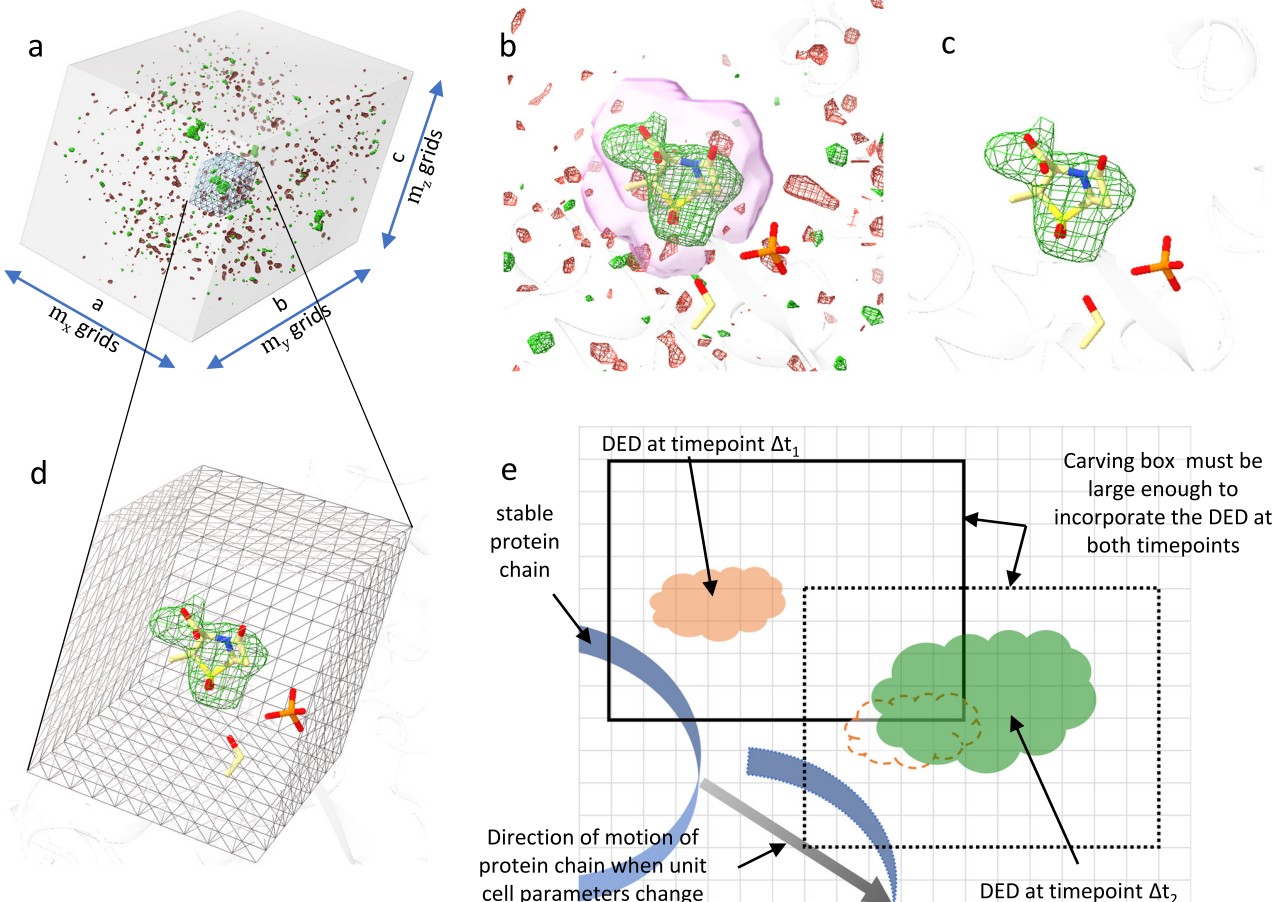

**Fig. 7 | Regions of interest (ROI) with changing unit cell parameters. a** DED map of the entire BlaC unit cell at $\Delta t_{misc} = 30$ ms contoured at ±2.5 σ **b** The same map as in **a** is now displayed with the focus on the active site of subunit A. Strong DED is present where the SUB molecule is located. A mask around the SUB atom is represented by a pink surface. The density inside the mask is left unaltered while that outside the mask is set to zero. **c** The map after the masking operation. **d** A box which is a part of overall map covers the ROI. **e** A simple 2D diagram showing how to choose the box. The square gray grid represents the voxels. The box shown by the solid line includes the ROI at $\Delta t_1$ and it is large enough to cover the evolving DED features at all other time points. The orange cloud represents the DED features at timepoint $\Delta t_1$. At $\Delta t_2$, the entire protein chain displaces to a new position (gray arrow) due to unit cell changes. In addition, more extensive DED features appear as shown by the green cloud. The dashed cloud next to the green DED is the relative position of the DED at $\Delta t_1$. As the protein chain displaces when the unit cell parameters change, the entire box moves accordingly in the same direction (shown by the dotted box). Since the number of grid points is adjusted linearly with the changing unit cell parameters, the number of voxels within the box as well as the voxel size do not change.

When the unit cell parameters do not change during the reaction, the difference maps at all time points will have the same number of voxels, and the voxel size is also constant. However, once the unit cell dimensions change, either the voxel size will change if the number of voxels is kept constant or the number of voxels will change if the voxel size is kept constant. If the voxel size changes, the DED value assigned to each voxel position will also change, which will skew the SVD analysis. If the voxel numbers change, the SVD algorithm will fail because it requires that all the maps are represented by arrays of identical sizes. Accordingly, both conditions, (i) a constant voxel size and (ii) a constant number of grid points in the masked volume, must be fulfilled when the unit cell changes.

Step 3: To fulfill (i), the total number of grid points in the DED map is changed proportionally to the unit cell change. When the volume of the ROI is not changed, condition (ii) is automatically fulfilled, and a suitable data matrix A can be constructed. However, when the unit cell parameters change, the ROI also changes position. This must be addressed in addition.

Step 4: A box is chosen that will cover the density that was just masked out (Fig. 7d). The box will include the ROI, which is saved as a new map. The size of the box must be large enough such that the ROIs can be covered at all time points. The box must be calculated with reference to a stable structure (usually the protein main chain). As the protein chain displaces as a result of the change in the unit cell, the box will also move accordingly to cover the correct ROI (Fig. 7e). As mentioned, the DED within the moving box can be used to evolve the mask that defines the final ROI, as indicated in step 2.

Step 5: All m voxels in the evolved mask are converted to a one-dimensional (1D) column array, a vector in high (m) dimensional space. How the conversion is achieved does not matter as long as the same convention is applied to all n maps. N of the m-dimensional vectors are arranged in ascending order of time to construct the data matrix A.

Step 6: SVD is performed on matrix A according to Eq. 1.

Step 7: Trial functions are globally fit to the significant rSVs to determine relaxation times and the minimum number of intermediates involved in the reaction (see, e.g., Ihee et al.[51]).

## Global fit of the significant rSVs

For a simple chemical kinetic mechanism with only first-order reactions, relaxations are characterized by simple exponentials[46]. For higher-order reactions, the rSVs must be fitted by suitable functions that must explain the changes in the electron density values in a chemically sensible way[15,16,112,113]. In our case, the significant rSVs were fitted by Eq. 2, which, apart from a constant term, consists of a logistic

function that accounts for abrupt changes in the electron densities observed in the active sites and an additional saturation term. Furthermore, the fit was weighted by the square of the corresponding singular values $S_i$.

$$S_i^2 rSV_i(t) = A_{0,i} + \frac{A_{1,i}}{1 + e^{-\lambda(t - \tau_1)}} + A_{2,i}(1 - e^{-\frac{t}{\tau_2}}) \qquad (2)$$

While the amplitudes A's are varied independently for each significant $rSV_i$ (i = 1…n), the parameter $\lambda$ and the relaxation times $\tau_1$ and $\tau_2$ are shared globally. The number of relaxation times (here, 2) is equal to the number of significant rSVs and to the number of distinguishable processes.

## Species concentrations

The diffusion of SUB molecules into the BlaC crystals and subsequently into the active sites triggers the reaction. Concentrations of SUB in the central flow (Supplementary Fig. 3) were estimated according to Calvey et al.[88]. At the longest $\Delta t_{misc}$ = 700 ms, the SUB concentration was 100 mM, which was used as the maximum ligand concentration for all calculations. The resulting evolution of the concentration of SUB at the active sites was modeled by Eq. 3.

$$I(t) = \frac{I_{max}}{1 + e^{-\mu(t - t_0)}} \qquad (3)$$

I(t) is the concentration of SUB at the active sites averaged over all unit cells in the crystal as a function of time, and $I_{max}$ is the maximum (100 mM) SUB concentration (Supplementary Table 4). Equation 3 is a logistic function which is a representation for an exponential growth with a limit. Initially the substrate concentration inside the crystal is zero and increases rapidly. However, the maximum outside concentration is 100 mM and the concentration quickly saturates to that value. The growth rate $\mu$ describes the steepness of the function. The parameter $t_0$ is a characteristic time point (in the middle of the transition) where the absolute growth reaches its maximum.

Once the SUB molecule reaches the active site of BlaC, the first step is the formation of a noncovalent enzyme inhibitor complex (E:I) (Fig. 6). The process depends on the free BlaC concentration inside the crystal and the rate coefficient for noncovalent complex formation ($k_{ncov}$). This step is usually reversibly defined by both the forward rate coefficient ($k_1$) and the backward rate coefficient ($k_{-1}$). However, the increasing concentrations of inhibitor inside the crystals force more molecules toward the active site. At least initially, the binding rate depends on $k_1$ alone. The noncovalent E:I complex is the reactant for the next phase of the reaction where the $\beta$-lactam ring opens. The resulting covalently bound acyl-enzyme complex (E-I) (Fig. 6) is so short-lived that it never accumulates in the timescale of the measurement. SUB undergoes rapid modification, and a product is formed where the enzyme is covalently bound to the irreversibly modified inhibitor (E- I*). $k_{cov}$ is the apparent rate coefficient that describes the velocity of E-I* formation directly from the E:I complex (Fig. 6). Ligand concentrations were determined by numerically integrating the following rate equations that describe the mechanism in Fig. 6.

$$\begin{aligned}
d[E:I] &= [E_{free}](t_i) \cdot [I](t_i) \cdot k_{ncov} \cdot dt \\
[E_{free}](t_{i+1}) &= [E_{free}](t_i) - d[E:I] \\
d[E-I^*] &= [E:I](t_i) \cdot k_{cov} \cdot dt \\
[E:I^*](t_{i+1}) &= [E-I^*](t_i) + d[E-I^*] \\
[E:I](t_{i+1}) &= [E:I](t_i) + d[E:I] - d[E-I^*] \\
t_{i+1} &= t_{i+1} + dt
\end{aligned} \qquad (4)$$

d[E:I] is the change in concentration of the noncovalent BlaC-SUB complex at any given time ($t_i$) and depends on the free enzyme concentration, $[E_{free}]$, the second-order rate coefficient of noncovalent binding ($k_{ncov}$), and the inhibitor concentration, [I]. [I] is calculated from Eq. 3. As the concentration of [E:I] increases, $[E_{free}]$ decreases. d[E-I*] is the increase in the covalently bound TEN. It depends on the available concentration of the noncovalent BlaC-SUB complex [E:I] and the rate coefficient $k_{cov}$. [E:I] decreases by the same rate [E- I*] increases.

The increase in the SUB concentration in the active site ($I_{in}$) is delayed relative to that of the SUB concentration in the unit cell ($I_{out}$). To account for this delay, the rate coefficient that determines the entry into the active site ($k_{entry}$, Fig. 2c) is assumed to be dependent on the concentration difference $\triangle I(t) = I_{out}(t) - I_{in}(t)$ between outside and inside the active site and a characteristic difference $\Delta I_c$. It is modeled by an exponential function

$$k_{entry} = k_{max,entry}\left(1 - \exp\left(-\frac{\Delta I(t)}{\Delta I_c}\right)\right) \qquad (5)$$

The relevant SUB concentrations within the active site ($I_{in}$) are generated by solving the following rate equation:

$$\begin{aligned}
dI_{in} &= k_{entry} \cdot I_{out} \cdot dt \\
I_{in}(t_{i+1}) &= I_{in}(t_i) + dI_{in}
\end{aligned} \qquad (6)$$

Equation 6 is used in lieu of Eq. 3 to calculate the relevant inhibitor concentration. $I_{in}$(t) is fed as [I](t) to Eq. 4 to calculate the concentrations of the noncovalently and covalently bound species shown in Fig. 5b. At early MISC delays, $k_{entry}$ is small. The channel opens, and $k_{entry}$ is large only when sufficient SUB has accumulated in the unit cell (Fig. 2d). All relevant parameters are listed in Table 1.

## Reporting summary

Further information on research design is available in the Nature Portfolio Reporting Summary linked to this article.

## Data availability

All relevant data are included with the paper and/or are available from the corresponding author upon reasonable request. The structure factors and the refined coordinates of the XFEL structure of BlaC mixed with sulbactam obtained after 3 ms, 6 ms, 15 ms, 30 ms, 240 ms, 700 ms and 3 h have been deposited in the Protein Data Bank (PDB) database under accession codes 8GCV, 8GCS, 8GCT, 8EBI, 8EBR, 8EC4, 8GCX and 8ECF respectively. Atomic coordinates of BlaC in other crystal forms used for comparison in this study are available in the PDB under accession codes 5OYO and 7A71. Source data are provided in this paper.

## Code availability

Code for each step of SVD calculation and the characterization of concentrations can be obtained from the authors upon request. These codes are available on zenodo[114](https://doi.org/10.5281/zenodo.8206588) and github (https://github.com/73km/pySVD4TX).

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

## Acknowledgements

This work was supported by NSF-STC-1231306 (BioXFEL). P.F. was supported by NSF BioXFEL STC grant NSF-1231306 Biology with X-ray Lasers, the NIH grant R01GM095583, and the ASU Biodesign Center for Applied Structural Discovery. A.O. was supported by the US Department of Energy, Office of Science, Basic Energy Sciences under award DE-SC0002164 (underlying dynamical techniques) and by the US National Science Foundation under awards STC-1231306 (underlying data analytical techniques) and DBI-2029533 (underlying analytical models). K.A.Z. was supported by the Cornell Molecular Biophysics Training Program (NIH T32-GM008267). D.F, L.A., and E.A.S. were supported by NSF STC BioXFEL center award 6227. L.A. training was supported in part by the National Institute of General Medical Sciences (NIGMS) of the National Institutes of Health (NIH) Maximizing Access to Research Careers (MARC) -T34 GM105549 grant. We acknowledge funding from DESY (Hamburg, Germany), a member of the Helmholtz Association HGF; the Cluster of Excellence "Advanced Imaging of Matter" of the Deutsche Forschungsgemeinschaft (DFG) - EXC 2056 - project ID 390715994; the Helmholtz Association Impulse and Networking fund - project InternLabs-0011 "HIR3X"; and the German Federal Ministry of Education and Research (BMBF) - project 05K18CHA. Use of the LCLS, SLAC National Accelerator Laboratory, is supported by the U.S. DOE, Office of Science, BES, under contract no. DE-AC02-76SF00515. The HERA system for in-helium experiments at MFX was developed by Bruce Doak and funded by the Max Planck Institute for Medical Research. One or more of the authors of this paper received support from a program designed to increase minority representation in science

## Author contributions

T.N.M., L.A., D.F., C.K., G.N.P., E.A.S., M.S., expressed, purified, and crystallized the protein. A. Batyuk, B.H., M.H., V.M., R.G.S., C.H.Y., Se.B. operated the SPB/SFX instrument. K.A.Z., L.P. designed and provided injector nozzles. A. Batyuk, B.H., M.H., S.L., R.G.S. assembled and operated the nozzles, T.N.M., Sa.B., B.H., M.H., C.K., S.L., V.M., R.G.S., C.H.Y., A. Barty., Se.B. collected the data. T.N.M., V.M., S.P., I.P., V.M., A.T., O.Y., C.H.Y., A. Barty., M.S. processed the data. T.N.M., S.P., I.P., and M.S. analyzed the data. T.N.M., M.S. wrote the pySVD4TX software. T.N.M., S.P., P.F., P.S., E.A.S, M.S. logged the experiment. T.N.M., J.K., J. M.-G., A.O., H.N.C., E.A.S., L.P., G.N.P., M.S. designed the experiment. T.N.M., M.S. wrote the manuscript with input from all other authors.

## Competing interests

The authors declare no competing interests.
