## [Peer Review File · Nature Communications]

Heterogeneity in the *M. tuberculosis* β -Lactamase Inhibition by SulbactamREVIEWER COMMENTS

Reviewer #1 (Remarks to the Author):

The authors describe the time-resolved, structural analysis binding of sulbactam to the beta-lactamase BlaC from *Mycobacterium tuberculosis*. It is impressive that such data sets can be obtained using serial crystallography. A new approach is described on how to analyze the data, which represent mixtures of states at different ratios at different delay times. That makes the manuscript rather technical. It also seems the major contribution of the work, making it more suitable for a crystallography journal.

The conclusions on the binding of sulbactam to BlaC, i.e. that induced fit and conformational selection play a role in reacting with sulbactam, are only valid for the peculiar crystal state, with four subunits in the asymmetric unit, two of which react slower and have a partly occluded active site due to the crystal organization. It is well established that BlaC is monomeric in solution, so such subunit interactions play no role under physiological conditions. Many other crystal structures of BlaC have two or even one molecule in the asymmetric unit, also suggesting that specific subunit interactions are solely a matter of crystal packing. The pH of the experiments is 4.1 - 4.5 which is probably also far from natural conditions inside the macrophages in which the bacterium lives. Thus, it would be wrong to conclude that BlaC in general works via these mechanisms without evidence from solution work.

As these findings most likely have no bearing on BlaC under more natural conditions, *Nature Communications* seems not an appropriate place for publication.

Reviewer #2 (Remarks to the Author):

The results of this paper show an incremental progression from the previous MISC studies of BlaC (Olmost et al., 2018 and Pandey et al., 2021) with an alternative substrate and reported increase in time-resolution. Additionally singular value decomposition with unit cell correction of the data is used to define the number of reaction intermediates and corresponding relaxation times. This provides evidence of kinetic heterogeneity in the various subunits within the asymmetric unit of the crystal, a useful consideration for time-resolved crystallography. Additionally, the development of the SVD method to allow for changes in unit cell during such experiments is reported.

Whilst this is of interest to the field the results are not particularly significant - the enzyme-inhibitor data at short time points prior to similar previously reported beta-lactam/inhibitor complexes is unconvincing. The data at all time points should be deposited to the protein databank. The work represents an incremental progression in previously reported methods and structural data.

The structural data presented shows some evidence of sublactam binding after 30 ms in subunit A and the trans-enamine intermediate is clearly shown at 15 ms in subunit B. The authors suggest there is evidence of the diffusion pathway of SUB into the active site indicated by changes in side-chain distances between guardian residues Ser70 and Arg173, Gln 112 and Gln 109. However, no density is shown to support this and the data presented do not conclusively show ligand binding or structural changes below 15 ms. To support discussion about sub 15 ms structural changes the data should be deposited in the PDB. The density shown around the ligand in video "Sublactam binding and reaction with subunit A of the BlaC" is ambiguous prior to 30 - 66 ms. Additionally the video "Active site opening in subunit A" should have indications of timing with supporting data deposited.

The constraint due to the diffusion of the inhibitor into the active site is an important consideration when determining the reported time resolution of the experiment. Additionally no significant changes are observed in the rSV of the data prior to the 15 ms time point.

The fit of equation 2 to the second rSV data in subunit D is not convincing and this should be discussed.

Minor issues:

Figure 4. ..The first and second significant rSV are shown by blue and red triangles respectively. Mislabeled the data-points are shown by squares.

Unit Cell of 6 ms dataset incorrectly quoted in b axis 80.4,8.8,112.1,108.7

In the supplementary information the authors state that Polder differences maps are used to display the ligand at some points particularly for the cryo-soaked data. This is a little ambiguous, in the main text only the cryo-soaked 3 hour time point for each subunit are labeled as a Polder difference map and this point should be clarified.

Whilst the methodology reported is sound the conclusions derived from the data are not fully supported by the current presentation.

The detail provided in the methods should allow the work to be reproduced when codes and data are made publicly available.

Reviewer #3 (Remarks to the Author):

The manuscript entitled “Heterogeneity in the *M. tuberculosis* β -Lactamase Inhibition by Sulbactam” by Malla et al. describes a time-resolved serial femtosecond crystallography study of the covalent binding of an inhibitor sulbactam (SUB) with the β -Lactamase BlaC. The authors performed their study with a mixing injector (MISC) whose efficiency has been proven in previous studies. A total of 8 structures have been produced: 7 time points ranging from 0 ms to 700 ms have been collected at MFX at LCLS and a 3h soak has been collected at ID-19 at APS.

The analysis of the difference of the electronic densities between the different time points combined with singular value decomposition (SVD) allowed the authors to explain how the inhibitor SUB binds to BlaC before being converted to trans-enamine (TEN) covalently bound to Ser70. The originality of this manuscript resides in the combination of serial crystallography with SVD using non-isomorphous dataset for the first time.

Major comments:

1) Binding heterogeneity between A/C and B/D is likely due to crystallographic artefact.

Lines 114-115: “Whereas the catalytic clefts of subunits B and D are wide open, those of subunits A and C are partially occluded by the neighboring subunits.”

For the subunits A and C, the binding of SUB is impaired by residues from subunits B and D, respectively. I would believe that what is observed in subunits A and C is not biologically relevant and is more an artefact due to crystal packing. However, subunit B and D do not show this issue and their behaviour is likely to be similar to what is happening in solution. The other structures of BlaC in the PDB, 6H2K (P1 space group) and 7A72 (P212121) seem to have accessible active site and no obstruction from other subunit in crystal packing.

Can the authors address this issue in the manuscript and comment why they chose this crystal form? Have they tried to grow slurry in different space group?

On a biological point of view in the discussion “Ligand Gating, Induced Fit and Conformational Selection” how relevant is the description of the 2-step mechanism for A/C inhibition?

2) SDV analysis

The quality of the analysis depends on the quality of the electronic density map. In this study the resolution is between 2.2 to 3.2 Å. Most of the critical time points have a resolution around 2.6-2.7 Å. How a low resolution and lower quality of the map (700 ms time point) influence the SVD analysis?

3) After reading the manuscript I feel like some experimental data are missing to corroborate SVD analysis. Predicted binding of TEN for subunits B and D is about 10 ms. A structure at 10 ms would validate the method SSX/ SVD. From the current data we know that the binding of SUB and the conversion to TEN happens in 9 ms (between 6 and 15 ms). I am not asking the authors to produce another structure since I understand that XFEL beamtime is hard to get and there is maybe no available nozzle for 10 ms time point. Would molecular dynamic simulation corroborate these findings? Would that be feasible in a reasonable time?

Minor comments:

4) Line 37: “we report in atomic detail...”, please include here “by room-temperature, time-resolved crystallography with timepoints between 3 and 700 ms , with diffraction data to between 2.2 and 2.7 Å resolutions.

5) Line 43: “cell parameters change...”, please indicate the magnitude of non-isomorphism and/or unit cell changes.

6) Lines 50-52, “In the last report ... security threat”: this sentence needs a reference which is called in the next sentence. Please move it here.

7) Line 69, add to the references 3,4,33: Rabe et al; Sci Adv. 2021 Aug 20;7(34):eabh0250. doi: 10.1126/sciadv.abh0250. Print 2021 Aug.

8) Lines 72-74; “...inhibitors with a β -lactam ring, irreversibly bind to β -lactamases and block their activity...” In fact, the beta lactam ring is no longer intact. It has been degraded but stuck in the active site. Could you please re-phrase it?

9) Line 85: It would be beneficial to add what is the quaternary structure of BlaC in solution. Is the enzyme active as a monomer, dimer, tetramer?

10) Lines 87-88: It would be a good place for adding a figure representing the lattice packing for several unit cells and solvent channels between them (see point #11, below)

11) Lines 91-92: In the PDB code 7K8K, it looks like in the sub-units A and C SUB is in the active site but bound in a non-productive manner (flipped), such that the atoms that should react are not properly oriented. Could you please comment?

12) Line 106, "...the unit cell parameters of the BlaC crystals vary after mixing...": please indicate the magnitude of unit cell change?

13) Line 112: "The alternating subunits are display..." should read "The alternating subunits are displaying..."

14) Lines 111-118, Binding site accessibility. The authors could try MAPS_CHANNELS to put number on the difference on the binding site accessibility between A/C and B/D.

Juers, D. H. & Ruffin, J. MAP_CHANNELS: a computation tool to aid in the visualization and characterization of solvent channels in macromolecular crystals. *J Appl Crystallogr* 47, 2105-2108, doi:10.1107/S160057671402281X (2014).

15) Line 119, "difference electron density (DED)". When authors mentioned difference electron density maps especially in figure 3, they should mention which maps are used. I suppose DED is isomorphous difference map in line 119 and figure 3? But sometimes it is for omit map. Could you please be more explicit about what kind of map is used.

16) Lines 119-120: for mixing experiment it is crucial to know the size of the crystal, temperature, SUB concentration and pH. Could you please add those details here?

17) Line 122: In general, please replace the abbreviation Tab. by Table

18) Line 125, "... positive densities ...": Is it Fo-Fo or Fo-Fc? I do not understand why SUB appears then disappears in subunit A? Were any of these timepoints data repeated?

19) Lines 125-126, "... spatially more spread out than that of water.": does the quality of the map at lowish resolution allows this interpretation? The electronic density map in figure 3 a is not very convincing. What are the B factors after refinement? Should you really place a SUB molecule in 3ms timepoint?

20) Line 133, "The β -lactam ring is oriented away from the Ser70. ": What the authors are describing here is a non-productive binding mode.

21) Line 138, "... were assembled to a movie...": If the authors are using morphing between time-resolved observations they should made it clear to the reader.

22) Lines 204-205, "While... different rates.": How biologically relevant it is? The reader may wonder if the enzyme is a monomer, dimer or tetramer.

23) Lines 207-208: "SUB displacements will be restricted by interactions with the surrounding residues (Extended Data Tab. 2).": do the authors mean the cavity is too restrictive to allow the full rotation of the non-productive into the productive binding mode? Perhaps they should state that the substrate may need to exit to allow for productive binding to occur?

24) Lines 209-211: It would be beneficial to have the cavity volume for subunits A/C vs subunits B/D.

25) Lines 215-216: "... a k_{cov} value of $\sim 1.5 \text{ mM}^{-1} \text{ s}^{-1}$ and a large k_{cov} value of $\sim 8000 \text{ s}^{-1}$ are estimated...": If possible, the authors should refer to and comment on other analysis measuring the rate of formation of covalent adduct for enzyme in solution.

26) Lines 221-222: "The two-step mechanism explains the enzymology in the active sites of all subunits in a consistent way.": It is likely that in solution, there is a possibility for productive and non-productive binding of SUB. the interconversion between these 2 states will have a rate that is probably impacted by the crystal lattice and exacerbated in subunits A/C.

27) Lines 234-235, "... the signature of an enzyme inhibitor complex is expected to appear already at the earliest time points (3 ms and 6 ms).": how is it consistent with the diffusion of SUB into crystals of 7 ms (mention above).

28) Lines 244-245, "...kncov (1.5 mM⁻¹ s⁻¹ for SUB compared to 3.2 mM⁻¹ s⁻¹ for CEF)...": the authors should be more explicit that these rates are based upon interpretation of sparse timepoint observations from time-resolved crystallography while steady state and/or transient kinetic analysis tend to be more data rich analysis.

29) Line 334-335: I agree with authors that current and upgraded synchrotrons will play a bigger role in time-resolved crystallography, but I really doubt that jet techniques like MISC are well suited for synchrotron source.

30) Line 380, Figure 1: k₁ and k₋₁ are inverted.

31) Line 380, Figure 1: If we consider the possibility of non-productive binding there are alternative kinetic models.

32) Line 382, Figure 2: In the legend there are two (b) and (d) is missing

33) Lines 411-412, "...73 could not be applied to X-ray data when large unit cell changes occur during the reaction...": Could the authors be more quantitative? How much is a large unit cell change?

34) Lines 463-473, Global fit of the Significant rSVs: It is not clear what is the justification of the logistic function. What is the biological justification for the logistic function to allow for burst of abrupt change and then followed by saturation that is apparently needed to fit the sparse data observations of the tr-SFX results.

35) Lines 482-483, "Eqn. 3 is a logistic function where μ is the growth rate and t_0 is the midpoint value of the growth." Same as point #31, could the authors expand on this for the general readers?

36) Lines 517-518, Extended Data table 1:

a) 700ms time point is at least 10 times less data than other time points with 2,854 hits indexed. Without seeing the electron density map it is hard to say, but is this timepoint good enough to be included in the analysis and especially using SVD?

b) Hits: please define what is a hit.

c) Missing units for rmsd bond length and bond angles

d) Occupancy of TEN decreased from 15 ms to 30 ms. Why?

e) Could you add the temperature of the data collection. 3h soak has been collected at cryo temperature while others at room temperature.

37) Line 521, Extended Data table 2: a figure with the atom numbering of SUB would be useful.

38) Line 537, Extended data figure 2:

a) (a) is it for subunit A? what is the resolution, contour, and colour scheme for the electronic density map? Should the reader assume that blue is 2Fo-Fc map?

b) (b) Which active site is it? What type of electronic density map?...

c) (d) State the resolution of the map.

d) (e)-(g): What is the resolution for the maps?

39) Line 542, "Extended data figure 3": (b) what is happening to all the 3 sigma features inside the mask? It is like removing unwanted density...

40) Line 553, Extended Data Figure 1: S atom colour in the reaction is too light for a white background. Please consider higher contrast.

We want to thank the reviewers for their careful and in-depth review of our manuscript. The manuscript has been revised based on their comments and suggestions as outlined in our point-to-point answers to the reviewer comments below.

To the best of our knowledge, kinetic data of sulbactam binding to *Mtb*. BlaC does not exist. This is the first study that investigates the binding kinetics of sulbactam, this time with time-resolved crystallography. We extract, by analyzing difference electron densities, kinetic parameters of sulbactam binding together with macromolecular structures.

We applied Singular Value Decomposition to identify the processes involved in the binding kinetics of SUB. Relaxation times from the SVD as well as occupancies of species involved provide a common picture of sulbactam binding that is heterogenous for two types of subunits. Questions asked with respect to the SVD analysis are addressed point by point. The programs involved are available on GitHub. Structures and structure factor amplitudes are deposited to the protein data bank.

REVIEWER COMMENTS

Reviewer #1 (Remarks to the Author):

The authors describe the time-resolved, structural analysis binding of sulbactam to the beta-lactamase BlaC from *Mycobacterium tuberculosis*. It is impressive that such data sets can be obtained using serial crystallography. A new approach is described on how to analyze the data, which represent mixtures of states at different ratios at different delay times. That makes the manuscript rather technical. It also seems the major contribution of the work, making it more suitable for a crystallography journal.

The conclusions on the binding of sulbactam to BlaC, i.e. that induced fit and conformational selection play a role in reacting with sulbactam, are only valid for the peculiar crystal state, with four subunits in the asymmetric unit, two of which react slower and have a partly occluded active site due to the crystal organization. It is well established that BlaC is monomeric in solution, so such subunit interactions play no role under physiological conditions. Many other crystal structures of BlaC have two or even one molecule in the asymmetric unit, also suggesting that specific subunit interactions are solely a matter of crystal packing. The pH of the experiments is 4.1 - 4.5 which is probably also far from natural conditions inside the macrophages in which the bacterium lives. Thus, it would be wrong to conclude that BlaC in general works via these mechanisms without evidence from solution work.

As these findings most likely have no bearing on BlaC under more natural conditions, Nature Communications seems not an appropriate place for publication.

We beg to differ with this reviewer on several points. The acid resistance in *Mycobacterium tuberculosis* is well studied. Numerous studies have shown that the pH of macrophage compartment where the *Mtb* bacterium resides is as low as 4.5 and more recently by Gouzy et. al [<https://doi.org/10.1073/pnas.2024571118>]. We changed the text and cited the publications. [Line 86 ff].

At pH 4.5 the BlaC is a dimer as determined by dynamic light scattering (see Olmos et al, 2018, supplementary material.). Analysis of the interface implemented in ChimeraX reveals that 30 residues are involved in the interface of subunits A and B of which 14 are from subunit A and 16 are from subunit B. Whereas only 23 residues are involved in the interface with subunits A and D. The A/B dimer is more likely.

The so called omega loop (residues 161-179) in β -lactamases play a significant role in substrate recognition and catalysis. The figure on the side shows subunit A of BlaC with the omega loop colored in orange. The catalytically active serine (SER70) and ARG173 on the tip of omega loop is also marked. Yi, H. et al. (2016) have demonstrated that the high adaptability of this omega loop is the foundation of extended spectrum β -lactamases that can hydrolyze large range of drugs by providing a flexible active site. Complemented by kinetic analysis in solution and using various mutants they showed the residues in this loop relaxed and regrouped upon substrate binding in an induced fit manner. Our findings are in agreement with their conclusion. The guardian residue ARG173 that lies on this omega loop is one of the residues closest to the active site. For a small ligand like sulbactam, the movement of this residue alone is sufficient to provide access to the active site. It is also important to note that the BlaC presented here is a wild type without any mutations. We have included this information in the text and cited corresponding references [Line 282-285].

Reviewer #2 (Remarks to the Author):

The results of this paper show an incremental progression from the previous MISC studies of BlaC (Olmost et al., 2018 and Pandey et al., 2021) with an alternative substrate and reported increase in time-resolution. Additionally singular value decomposition with unit cell correction of the data is used to define the number of reaction intermediates and corresponding relaxation times. This provides evidence of kinetic heterogeneity in the various subunits within the asymmetric unit of the crystal, a useful consideration for time-resolved crystallography.

Additionally, the development of the SVD method to allow for changes in unit cell during such experiments is reported.

Whilst this is of interest to the field the results are not particularly significant - the enzyme-inhibitor data at short time points prior to similar previously reported beta-lactam/inhibitor complexes is unconvincing. The data at all time points should be deposited to the protein databank. The work represents an incremental progression in previously reported methods and structural data.

The structural data for all time points have now been deposited into the PDB data bank 8GCV, 8GCS, 8GCT, 8EBI, 8EBR, 8EC4, 8GCX and 8ECF (see 'data availability'). Our study presents a kinetic analysis of mix-and-inject crystallographic data collected at an XFEL source. In addition, the SVD algorithm is rewritten entirely to accommodate any variation in unit cells of crystallographic data. SVD informed kinetics and the observation of difference electron density made it possible that time dependent concentrations of enzyme and ligand inside the crystals can be estimated. These are all novel techniques of high general relevance that accurately extend the analysis of single states in classical crystallography to unbiased estimates of mixtures of states. These methods can be applied not only to MISC data but to all other time-resolved X-ray data.

The structural data presented shows some evidence of sublactam binding after 30 ms in subunit A and the trans-enamine intermediate is clearly shown at 15 ms in subunit B. The authors suggest there is evidence of the diffusion pathway of SUB into the active site indicated by changes in side-chain distances between guardian residues Ser70 and Arg173, Gln 112 and Gln 109. However, no density is shown to support this and the data presented do not conclusively show ligand binding or structural changes below 15 ms. To support discussion about sub 15 ms structural changes the data should be deposited in the PDB.

We submitted the 3 ms and 6 ms data to the pdb together with refined structures that do not contain ligands in subunit A and B but they contain the mentioned side chain positional changes, so that we could calculate distances. As clarified in the text (line 129 ff), some (weak) extra density is observed at the entrance, which is not clear enough to model and refine a substrate molecule.

The changes in distances mentioned in the paper are calculated from the refined structures which are ultimately derived from the experimental data. It would have been impossible to show maps for each data set and the changes were summarized into a table for clarity and easy comparison. For example, the image below shows 2FO-FC map contoured at 1σ for unmixed (in yellow) and 15ms (in purple) datasets. The map focuses on the entrance to the active site in subunit A. The guardian residues (GLN 112 and ARG 173) are shown. The GLN 112 side chain has drastically turned in the opposite direction and ARG 173 has moved backward from the initial position. Evidence of these structural changes led us to believe the diffusion of

sulbactam into the active site is in progress even though a sulbactam cannot be modeled at these early delays.

The density shown around the ligand in video "Sulbactam binding and reaction with subunit A of the BlaC" is ambiguous prior to 30 - 66 ms. Additionally the video "Active site opening in subunit A" should have indications of timing with supporting data deposited.

We completely agree, before 30 ms (at 3 ms and 6 ms) the density is ambiguous. Therefore, we did not refine a ligand. However, at 30 ms (and later time points), a ligand be placed unambiguously. Before 30 ms in subunits A/C, the displacements of the surrounding residues are clearly visible which we believe is induced by the appearance of SUB near the active site. The SUB itself cannot be modeled with confidence as the densities are weak. This may be due to the translational and rotational disorder of the free SUB as it moves closer to the reaction site at early time scales. These weak densities were not included in the movie making but they are still included in the main paper in Figure 3.

The active site opening video was created to show the extent of opening of the entrance. It has now been updated with time stamps.

The constraint due to the diffusion of the inhibitor into the active site is an important consideration when determining the reported time resolution of the experiment.

To increase the ligand concentration rapidly in the crystals, we add SUB at a concentration of 150 mM. This means that stoichiometric concentrations are reached on a few ms time scale after mixing. It is important to point this out, since in our case (when the substrate concentration is much higher than the protein concentration in the crystal) diffusion times into the crystals are much longer than the time required to reach stoichiometric concentrations. Already at the earliest time-points sufficient ligand is available. The reason that we do not

observe ligand in the active site is that binding of SUB to both subunits is rather slow compared to the buildup of SUB in the unit cell. Diffusion is surprisingly rapid when sufficient ligand concentration is available. A similar scenario is pointed out in the Pandey et 2021 paper and also mentioned in the Crystals paper (Schmidt 2020). The text is amended and both papers are cited.

Additionally no significant changes are observed in the rSV of the data prior to the 15 ms time point.

The rSVs are the direct result of the electron density present at the given time points. The electron densities at the earliest timepoints are very weak and we did not interpret them. Accordingly, they also do not show significant changes in the rSVs.

The fit of equation 2 to the second rSV data in subunit D is not convincing and this should be discussed.

To perform time resolved analysis in an ideal scenario, all the datasets are collected in an identical way and have same the quality and resolution. Here, all but the 66 ms dataset were collected together at the same beamtime. The 66ms dataset was collected at a different beamtime, at a different X-ray facility (EuXFEL) with different conditions e.g. a high-repetition rate pulse structure, with a different X-ray energy (9.2 keV at EuXFEL vs. 9.8 keV at LCLS), a different X-ray detector (AGIPD at EuXFEL versus the ePix-10k at LCLS) and using a different BlaC microcrystal preparation. Since SVD is very sensitive to variations in data collection conditions, the data point in the 2nd rSV for the 66 ms dataset in subunit D appears to be a slight outlier. The trans-enamine is already formed in this subunit at 15ms and no big structural changes are apparent afterwards. There is no second process to produce a strong signal in the 2nd rSV. Upon omission of this data, the fit is much better without impacting the kinetic parameters that were determined already. We include a footnote in the caption of the figures 4 and 5 mentioning the 66 ms data is obtained from a different experiment.

Minor issues:

Figure 4. ..The first and second significant rSV are shown by blue and red triangles respectively. Mislabeled the data-points are shown by squares.

We corrected the label appropriately.

Unit Cell of 6 ms dataset incorrectly quoted in b axis 80.4,8.8,112.1,108.7

We corrected the table.

In the supplementary information the authors state that Polder differences maps are used to display the ligand at some points particularly for the cryo-soaked data. This is a little

ambiguous, in the main text only the cryo-soaked 3 hour time point for each subunit are labeled as a Polder difference map and this point should be clarified.

The label on Figure 3 has been updated and now includes the following text: “... Omit map is shown in all panels except for panels d, f, j and l which are shown in Polder maps.” This means polder maps were calculated for the 30 ms and 700 ms data from the XFEL.

Whilst the methodology reported is sound the conclusions derived from the data are not fully supported by the current presentation.

With mix-and-inject serial crystallography, the transient enzyme/ligand complexes engaged in enzymatic reactions in millisecond timescales can be visualized at near atomic resolution. The enzyme is fully functional at physiological temperature and does not require any mutation to trap intermediates as has been done previously [DOI: 10.1021/ja3073676].

The conclusion that a ligand with multiple reactive moieties might enhance the efficacy of the drugs is not a stand-alone conclusion but is supported by independent evidence in this currently very active research field [Mohamad et al., 2022, DOI: 10.1021/acsomega.2c05212, White et al., 2019, DOI: 10.1016/j.bmcl.2019.07.002]. It has been shown that the molecular structure of β -lactam compounds can be modified by adding another β -lactam ring or other pharmacophores that modify their pharmacological properties and spectrum of action [De Rosa et al., 2021, <https://doi.org/10.3390/ijms22020617>]. The authors of these publications suggest a drug that mimics the combination between an antibiotic and a β -lactamase inhibitor.

The induced fit mechanism of catalysis by β -lactamases is already established. The ability of this enzyme to adapt to the wide range of substrates varying in size by relaxing the active site to accommodate them and regroup upon binding is linked to the extended spectrum of antibiotic resistance [Yi et al., 2016, doi: 10.1038/srep36527].

Yi et al., 2016 and De Rosa et al 2021 are cited in the text as they contain the appropriate information.

We believe, the novel techniques introduced here, which enable researchers to extract the kinetics of the reaction directly from time resolved X-ray data alone might become useful to investigate other enzymatic reactions at XFELs and synchrotrons.

The detail provided in the methods should allow the work to be reproduced when codes and data are made publicly available.

We put the code for the SVD analysis on GitHub [<https://github.com/73km/pySVD4TX>]

Reviewer #3 (Remarks to the Author):

The manuscript entitled “Heterogeneity in the M. tuberculosis β -Lactamase Inhibition by

Sulbactam” by Malla et al. describes a time-resolved serial femtosecond crystallography study of the covalent binding of an inhibitor sulbactam (SUB) with the β -Lactamase BlaC. The authors performed their study with a mixing injector (MISC) whose efficiency has been proven in previous studies. A total of 8 structures have been produced: 7 time points ranging from 0 ms to 700 ms have been collected at MFX at LCLS and a 3h soak has been collected at ID-19 at APS. The analysis of the difference of the electronic densities between the different time points combined with singular value decomposition (SVD) allowed the authors to explain how the inhibitor SUB binds to BlaC before being converted to trans-enamine (TEN) covalently bound to Ser70. The originality of this manuscript resides in the combination of serial crystallography with SVD using non-isomorphous dataset for the first time.

Major comments:

1) Binding heterogeneity between A/C and B/D is likely due to crystallographic artefact. Lines 114-115: “Whereas the catalytic clefts of subunits B and D are wide open, those of subunits A and C are partially occluded by the neighboring subunits.”

Many thanks for pointing this out.

For the subunits A and C, the binding of SUB is impaired by residues from subunits B and D, respectively. I would believe that what is observed in subunits A and C is not biologically relevant and is more an artefact due to crystal packing. However, subunit B and D do not show this issue and their behaviour is likely to be similar to what is happening in solution. The other structures of BlaC in the PDB, 6H2K (P1 space group) and 7A72 (P212121) seem to have accessible active site and no obstruction from other subunit in crystal packing.

Can the authors address this issue in the manuscript and comment why they chose this crystal form? Have they tried to grow slurry in different space group?

There is no time resolved solution study performed on the sulbactam reaction with Mtb beta lactamase for direct comparison.

In our previous experiment, Olmos et al. 2018 [<https://doi.org/10.1186/s12915-018-0524-5>], we had two types of crystal forms (compare 6A6B and 6B5X). One with a monomer in the unit cell and another which is the same one as presented in this manuscript. In monomeric forms like 7A72, 6A6B and even dimeric forms like 6H2K found in the literature, the molecules are more densely packed. The diffusion relies on the transient opening and closing of channels between the molecules. However, in the crystal form used here there are solvent channels as wide as 30Å which allows the ligand to traverse through the unit cell (and thereby the crystal) in all directions. The essentially free diffusion ensures swift and uniform reaction initiation. The text is updated on line 86 ff.

On a biological point of view in the discussion “Ligand Gating, Induced Fit and Conformational Selection” how relevant is the description of the 2-step mechanism for A/C inhibition?

The two-step inhibition mechanism is equally valid for A/C inhibition. In fact, the effect is more pronounced here. The first step has rapid association-dissociation kinetics. But due to the narrow cavity immediate dissociation is hindered once a complex is formed. The transformation of non-covalent complex to covalent one is also slowed down. As a result, the distinct intermediate formed in the first step can be observed. It is because of the unique nature of the active site such intermediate in the pre-acylation stage could be trapped on a wild type enzyme. Previous efforts to trap this non-covalent intermediate were possible only with point residue mutation of the catalytically active Ser70. However, with this mutation the reaction could never reach completion [Rodkey et al., 2012, DOI: 10.1021/ja3073676]. In the second step, the inhibitor binding induces a conformational change in the enzyme, resulting in stabilization of the enzyme-inhibitor complex. Without such stable complex, prolonged inhibition is not possible. This is well reviewed in Blat et al 2010 [<https://doi.org/10.1111/j.1747-0285.2010.00972.x>]. The Rodkey et al publication is cited (line 93).

The 2-step mechanism is not exclusive to BlaC. Many reactions catalyzed by enzymes are described by both a 2-step mechanism and by an induced fit [Fieuline, et al. 2011, <https://doi.org/10.1371/journal.pbio.1001066>, Tumino and Copeland, 2008, <https://doi.org/10.1021/bi8002023>]. We included this information in the text on line 207.

2) SDV analysis

The quality of the analysis depends on the quality of the electronic density map. In this study the resolution is between 2.2 to 3.2 Å. Most of the critical time points have a resolution around 2.6-2.7 Å. How a low resolution and lower quality of the map (700 ms time point) influence the SVD analysis?

SVD appears to robust against variations in the number of Diffraction Patterns. Even though the resolution is lower than other time points, there is presence of strong electron density of the ligand. Also see the image on comment 36. The signal is clearly distinguishable from the background noise. These qualities made it possible to include 700 ms in the SVD analysis. And as evident from the rSV graphs, the 700 ms data set is consistent with the overall analysis. It should be noted that the major structural relaxations have occurred by 240 ms and there is little change in the signal between 240 ms and 700 ms.

3) After reading the manuscript I feel like some experimental data are missing to corroborate SVD analysis. Predicted binding of TEN for subunits B and D is about 10 ms. A structure at 10 ms would validate the method SSX/ SVD. From the current data we know that the binding of SUB and the conversion to TEN happens in 9 ms (between 6 and 15 ms). I am not asking the authors

to produce another structure since I understand that XFEL beamtime is hard to get and there is maybe no available nozzle for 10 ms time point. Would molecular dynamic simulation corroborate these findings? Would that be feasible in a reasonable time?

Indeed, we agree with the reviewer that the XFEL beamtimes are hard to get, so the collection of more data points is not feasible. A 10 ms time-point would shift the relaxation time measured in the SVD analysis by about 2 ms. This timing accuracy is difficult to achieve in a molecular dynamics simulation where we expect that the timing inaccuracy is much larger than the said 2 ms given the technical problem to perform an all-atom MD, which is further complicated by the fact that it would have to simulate the environment of a crystal including diffusion on the millisecond time scale.

Minor comments:

4) Line 37: “we report in atomic detail...”, please include here “by room-temperature, time-resolved crystallography with timepoints between 3 ms and 700 ms , with diffraction data to between 2.2 and 2.7 Å resolutions.

Agreed, we included this in the abstract.

5) Line 43: “cell parameters change...”, please indicate the magnitude of non-isomorphism and/or unit cell changes.

We have changed the text. Now it reads “... remains functional even if unit cell parameters change up to 3 Å during the reaction.” (line 44)

6) Lines 50-52, “In the last report ... security threat”: this sentence needs a reference which is called in the next sentence. Please move it here.

We moved the reference.

7) Line 69, add to the references 3,4,33: Rabe et al; Sci Adv. 2021 Aug 20;7(34):eabh0250. doi: 10.1126/sciadv.abh0250. Print 2021 Aug.

We have added the reference.

8) Lines 72-74; “...inhibitors with a β -lactam ring, irreversibly bind to β -lactamases and block their activity...” In fact, the beta lactam ring is no longer intact. It has been degraded but stuck in the active site. Could you please re-phrase it?

We rephrased the sentence to “... Sulbactam (SUB), clavulanate and tazobactam, all β -lactam based suicide inhibitors, irreversibly bind to β -lactamases and block their activity...” (line 74)

9) Line 85: It would be beneficial to add what is the quaternary structure of BlaC in solution. Is the enzyme active as a monomer, dimer, tetramer?

In our previous paper (Olmos et al., 2018) we present an analysis performed by dynamic light scattering that shows that the BlaC is a dimer at low pH. We comment on this in the text on line 86 ff.

10) Lines 87-88: It would be a good place for adding a figure representing the lattice packing for several unit cells and solvent channels between them (see point #11, below)

This has been done in the previously published paper where the reaction of BlaC with ceftriaxone was investigated. The following figure is taken from Supplementary file of Olmos et al. 2018 [<https://doi.org/10.1186/s12915-018-0524-5>]. We explicitly refer to this figure now in the text (line 91).

11) Lines 91-92: In the PDB code 7K8K, it looks like in the sub-units A and C SUB is in the active site but bound in a non-productive manner (flipped), such that the atoms that should react are not properly oriented. Could you please comment?

This is exactly what we observe, and our discussions for subunits A and C are based on this [Line 205-214]. Initially, a non-productive complex is formed. However, at longer time points a fully reacted trans-enamine is found instead of a non-covalently bound intact sulbactam. The reaction

is successfully completed, however later than in subunits B and D. An authentic enzyme-inhibitor complex is formed.

12) Line 106, "...the unit cell parameters of the BlaC crystals vary after mixing...": please indicate the magnitude of unit cell change?

The sentence has been modified to include "...the unit cell parameters of the BlaC crystals vary up to 3 Å after mixing..."

13) Line 112: "The alternating subunits are display..." should read "The alternating subunits are displaying..."

We corrected the text.

14) Lines 111-118, Biding site accessibility. The authors could try MAPS_CHANNELS to put number on the difference on the binding site accessibility between A/C and B/D.

Juers, D. H. & Ruffin, J. MAP_CHANNELS: a computation tool to aid in the visualization and characterization of solvent channels in macromolecular crystals. J Appl Crystallogr 47, 2105-2108, doi:10.1107/S160057671402281X (2014).

We thank the reviewer for pointing to this resource. We calculated the solvent accessible channel for reference structure with the probe radius equivalent to that of SUB molecule (3.5 Å). The output is shown in the image below. Van der Waal surface is shown for the protein in khaki and the solvent channel in transparent blue. There is only one contiguous channel traversing the unit cell in all three directions (Fig a). The SUB accessible channel extends close to, but does not penetrate the active site in subunit A whose location is marked by the nearby residing phosphate molecule (Fig b). The channel remains contiguous until the cutoff radius of 15 Å which is consistent with the 30 Å large opening in the structure. We changed the text to include this information and added the citation [line 91-92].

15) Line 119, "difference electron density (DED)". When authors mentioned difference electron

density maps especially in figure 3, they should mention which maps are used. I suppose DED is isomorphous difference map in line 119 and figure 3? But sometimes it is for omit map. Could you please be more explicit about what kind of map is used.

Isomorphous difference ($F_o - F_c$) maps do not show appropriate signal. We had to calculate $F_o - F_c$ maps. This is due to the non-isomorphism caused by large unit cell changes. A large fraction of the paper deals with methods to analyze these types of difference maps. This is new. We have added the following sentence in line 126 to clarify this. "... As the datasets are non-isomorphous, simulated annealing $mF_o - DF_c$ omit maps and polder maps were calculated as appropriate. ..." In addition, the caption in Fig. 3 is also updated to indicate which maps from XFEL data are presented with omit and which by polder maps.

16) Lines 119-120: for mixing experiment it is crucial to know the size of the crystal, temperature, SUB concentration and pH. Could you please add those details here?

SUB is provided in sodium phosphate buffer (50 mM) at pH 4.5. The details on mixing parameters are presented in supplementary Table 1. The crystals size ($\sim 10 \times 10 \times 2 \mu\text{m}^3$) is reported in in the Supplementary text [Line 27].

17) Line 122: In general, please replace the abbreviation Tab. by Table

We have replaced all the abbreviated Tab. by Table.

18) Line 125, "... positive densities ...": Is it $F_o - F_c$ or $F_o - F_o$? I do not understand why SUB appears then disappears in subunit A? Were any of these timepoints data repeated?

All maps presented in this manuscript are either $mF_o - DF_c$ omit maps or Polder maps. At 3ms the density is *outside* the entrance of the active site. As the entrance opens at 6 ms and 15 ms, the SUB moves closer to the reaction site and the density *outside* the entrance diminishes. Once the SUB is completely *inside* the active site at and after 30 ms, the density persists and transforms eventually to trans-enamine at longer time points.

19) Lines 125-126, "... spatially more spread out than that of water.": does the quality of the map at lowish resolution allows this interpretation? The electronic density map in figure 3 a is not very convincing. What are the B factors after refinement? Should you really place a SUB molecule in 3 ms timepoint?

We agree with the reviewer and we pointed this out in the text [Line 131-133]. We believe the density *outside* the active site at 3ms is likely caused by the accumulation of SUB before it enters inside. We only placed the SUB molecule there for representation (as a guide to the eye), but it was not *refined*. It is addressed in line 133 as quote: "... However, refinement of the SUB is difficult (see supplementary Fig. 1 a) ...". The caption on Supplementary Fig. 1a reads "SUB

(shown in purple) was placed in the DED_{omit} feature near the guardian residues observed at $\Delta t_{misc} = 3$ ms (Fig. 3 a) and refined. After refinement negative density (red contour lines) appear in the $F_{obs} - F_{calc}$ map (displayed on the $\pm 3\sigma$ level). The B-factors of all SUB atoms are in excess of 120 \AA^2 . Translational and rotational disorder of the free SUB near the guardian residues renders the refinement of a single conformation difficult...”

20) Line 133, “The β -lactam ring is oriented away from the Ser70. “: What the authors are describing here is a non-productive binding mode.

SUB in subunits A/C eventually reacts with the BlaC to a covalently bound trans-enamine. We agree with the reviewer that some fraction of the binding could initially be non-productive and additional steps could be required that will lead to same end-result. We generalized the reaction into a much simpler 2-step mechanism as shown in Fig. 1. The initial binding is considered an enzyme-inhibitor complex formation

21) Line 138, “... were assembled to a movie...”: If the authors are using morphing between time-resolved observations they should made it clear to the reader.

It is now mentioned and updated in the supplementary information guide as “... The movie has been assembled by interpolation (morphing) between the measured structures at 3 ms, 6 ms, 15 ms, 30 ms, 66 ms, 240 ms and 700 ms. Trajectories between the measured species are, therefore, artificial and should be regarded with caution as a guide to the eye...”

22) Lines 204-205, “While... different rates.”: How biologically relevant it is? The reader may wonder if the enzyme is a monomer, dimer or tetramer.

At pH 4.5 the BlaC is a dimer as determined by dynamic light scattering (see Olmos et al, 2018, supplementary material.). As pointed out above, the analysis of the interface implemented in ChimeraX reveals that 30 residues are involved in the interface of subunits A and B of which 14 are from subunit A and 16 are from subunit B. Whereas 23 residues are only involved in the interface with subunit D. The A/B dimer is more likely.

23) Lines 207-208: “SUB displacements will be restricted by interactions with the surrounding residues (Extended Data Tab. 2).”: do the authors mean the cavity is too restrictive to allow the full rotation of the non-productive into the productive binding mode? Perhaps they should state that the substrate may need to exit to allow for productive binding to occur?

We do not think that the cavity is too restrictive to allow for rotation of the non-covalently bound SUB into the productive binding mode. Our model is that SUB binds only effectively and

non-covalently in the up-side-down (non-productive) orientation. The SUB will then rotate in the active site by overcoming an energy barrier to the productive state. However, the idea that there is an equilibrium between an unproductive orientation and a productive orientation is appealing. An alternative model would be that SUB would preferentially bind in the non-productive state but through exit of the unproductive and re-entry of a productive orientation the reaction would proceed to the trans-enamine. Rather than rotating within the cavity, rotation would happen outside, or a completely new molecule would enter in the correct orientation. However, the entry rate of the SUB in its “putative” productive orientation must be extremely small, otherwise the trans-enamine would accumulate much earlier than observed. We believe this is unlikely.

24) Lines 209-211: It would be beneficial to have the cavity volume for subunits A/C vs subunits B/D.

We used the Computed Atlas of Surface Topography of proteins (CASTp) analysis to calculate the volume of the active site. The figure generated by the CASTp server is shown below. In the figure, the blue surface represents the enclosed volume at the active site in subunit A and yellow is for subunit B. The solvent accessible volume of the active site in subunit A is calculated to be $\sim 93 \text{ \AA}^3$. As the active site of subunit B is wide open, an accurate volume cannot be calculated. Still, the program reports the volume for subunit B to be $\sim 2621 \text{ \AA}^3$ which includes the entire volume in the center of the asymmetric unit. We included this information in the text and cited the CASTp paper. [line 209-212]

25) Lines 215-216: “... a k_{cov} value of $\sim 1.5 \text{ mM}^{-1} \text{ s}^{-1}$ and a large k_{cov} value of $\sim 8000 \text{ s}^{-1}$ are estimated...”: If possible, the authors should refer to and comment on other analysis measuring the rate of formation of covalent adduct for enzyme in solution.

To the best of our knowledge, we found no literature available for the kinetic analysis of sulbactam reaction with *Mtb* β -lactamase. The kinetic parameters of sulbactam reacting with

other β -lactamases determined by initial rate measurement technique varied widely even among the same class of β -lactamases. We included this in the main text. [Line 225-230]

26) Lines 221-222: “The two-step mechanism explains the enzymology in the active sites of all subunits in a consistent way.”: It is likely that in solution, there is a possibility for productive and non-productive binding of SUB. the interconversion between these 2 states will have a rate that is probably impacted by the crystal lattice and exacerbated in subunits A/C.

We agree with the reviewer about the possibility. The intact sulbactam observed in subunits A/C is an important intermediate on the pre-acylation stage. Without the slowly reacting active site, such observation would have been impossible as evident in subunits B/D. On longer time delays, the reaction proceeds to completion by forming covalent bond. In other experiments, such intermediate has been trapped with impaired enzymes where the catalytically active Ser70 is mutated such that the intact sulbactam accumulates but never reacts.

27) Lines 234-235, “... the signature of an enzyme inhibitor complex is expected to appear already at the earliest time points (3 ms and 6 ms).”: how is it consistent with the diffusion of SUB into crystals of 7 ms (mention above).

The concentration of BlaC molecules in the crystal is ~ 16 mM which is mixed with 150 mM SUB solution. Stoichiometric concentration of ligand (1:1 ratio of ligand molecules to active sites) is reached already at 4 - 5 ms after mixing (Fig. 5) . At 7 ms, the concentration increases to ~ 50 mM which is greater than 3 times the concentration of BlaC molecules. We added this information in the text [Line 250, Supplementary Information line 30-34].

28) Lines 244-245, “...kncov ($1.5 \text{ mM}^{-1} \text{ s}^{-1}$ for SUB compared to $3.2 \text{ mM}^{-1} \text{ s}^{-1}$ for CEF)...”: the authors should be more explicit that these rates are based upon interpretation of sparse timepoint observations from time-resolved crystallography while steady state and/or transient kinetic analysis tend to be more data rich analysis.

We agree, but with a caveat. There are several reasons why to use crystallography to determine kinetic mechanisms (the kinetics). (1) the species structures (and their difference electron densities) are sometime obvious even with only a few time points. (2) In crystallography, concentration is directly related to electron density. In other methods there is a linear factor (for example the wavelength dependent extinction coefficients) that complicates the analysis. (3) We hope (expect) that in the future when high-repetition rate XFELs become available at high energy (the LCLS II high energy upgrade is planned for 2028-30) many more time points can be collected during an allocated beamtime. The methods presented here will be of importance for the analysis of these data. (4) Kinetics is mostly exponential. This means that time-points must be spread out linearly as a function of log-time. From simulations conducted previously for the paper that introduces the application of the singular value decomposition to X-ray data (Schmidt et al.,

2003), we reasoned that 2 to 3 time-points per logarithmic decade are usually sufficient to determine candidates of mechanisms. In our study we spread out the time-points as much as we could on log-time to cover the time range from 3 ms to 700 ms. We amended the text in line 346 ff.

29) Line 334-335: I agree with authors that current and upgraded synchrotrons will play a bigger role in time-resolved crystallography, but I really doubt that jet techniques like MISC are well suited for synchrotron source.

Agreed, other methods such as the fixed target or a conveyor belt have to be employed at the synchrotron. New beamlines have been built (for example at ESRF ID29 SMX beamline) that has just started operations which allows for serial millisecond crystallography data to be collected with up to 1000 images per second and a photon flux /shot corresponding to 1/10 of the single shot intensity at LCLS. These new beamlines may in the future allow for TR-SMX mix and inject data collection at Synchrotrons. We updated the text and added relevant references (Mehrabi et al., 2019, Beyerlein et al., 2017, Roessler et al., 2016 [Line 348].

30) Line 380, Figure 1: k1 and k-1 are inverted.

The figure is corrected.

31) Line 380, Figure 1: If we consider the possibility of non-productive binding there are alternative kinetic models.

We agree with the reviewer. There might be more pathways of this reaction. This is a simplified model which is sufficient to explain the observations from the data sets. We comment on this in the caption of Fig. 1.

32) Line 382, Figure 2: In the legend there are two (b) and (d) is missing

The caption is corrected.

33) Lines 411-412, "...73 could not be applied to X-ray data when large unit cell changes occur during the reaction...": Could the authors be more quantitative? How much is a large unit cell change?

Typically, in crystallography unit cell changes larger than 1/4 of max resolution are considered non-isomorphous (see J. Drenth, Principles of Protein Crystallography, 2nd edition, p. 139). We observed unit cell changes larger than 0.6 Å (and up to 3 Å) with resolution limits of around 2.4 Å. As a result, the difference signal in Fobs(t)-Fobs(reference) difference maps is greatly affected. We updated the text to include this information. [Line 377]

34) Lines 463-473, Global fit of the Significant rSVs: It is not clear what is the justification of the logistic function. What is the biological justification for the logistic function to allow for burst of abrupt change and then followed by saturation that is apparently needed to fit the sparse data observations of the tr-SFX results.

The logistic function is a convenient representation of a sigmoidal saturation curve. It describes exponential growth with a limit after a lag phase. Here, the logistic function is used to describe the diffusion of the ligand molecules inside the crystal. Initially the substrate concentration inside the crystal is zero and increases rapidly. However, the maximum outside concentration is 100 mM and the concentration quickly saturates to that value. The increase in ligand concentration inside the crystal is thus logistic. This is then reflected in the nature of rSVs derived from the experimental data.

35) Lines 482-483, "Eqn. 3 is a logistic function where μ is the growth rate and t_0 is the midpoint value of the growth." Same as point #31, could the authors expand on this for the general readers?

The logistics function is a representation of a step-function with exponential growth. The parameters of a logistic equation describe where the growth starts and where it saturates. The growth rate represents the steepness of the function. In between these parts is a characteristic time point where the growth reaches its maximum. We have added this to the text. [Line 446-451]

36) Lines 517-518, Extended Data table 1:

- a) 700ms time point is a least 10 time less data than other time point with 2,854 hits indexed. Without seeing the electron density map it is hard to say, but is this timepoint good enough to be included in the analysis and especially using SVD?
- b) Hits: please define what is a hit.
- c) Missing units for rmsd bond length and bond angles
- d) Occupancy of TEN decreased from 15 ms to 30 ms. Why?
- e) Could you add the temperature of the data collection. 3h soak has been collected at cryo temperature while others at room temperature.

(a) The picture below shows the Omit maps (*not polder*) at the active site of subunits A-D respectively for 700ms dataset. Due to low number of indexed patterns, the density is weaker and had to be shown on lower contour level which are marked for each subunit separately. However, there is clear distinction of the signal from the background. This made it possible to include the data in the SVD analysis

(b) A hit is an expression that indicates Bragg reflections found in the diffraction patterns. We included a footnote in supplementary table 1. We apologize for using this expression, as it sounds like being lab-jargon. But it is used widely in the SFX community.

(c) We added the units.

(d) We consider this slight decline as the error in our occupancy refinement.

(e) The temperature parameter has been added to the data collection table.

37) Line 521, Extended Data table 2: a figure with the atom numbering of SUB would be useful.

This information is located in Fig. 1. Fig. 1b has been updated with atom numbering for sulbactam and trans-enamine.

38) Line 537, Extended data figure 2:

a) (a) is it for subunit A? what is the resolution, contour, and colour scheme for the electronic density map? Should the reader assume that blue is 2Fo-Fc map?

This is subunit A. The resolution is 2.6 Å. Blue contour map is 2Fo-Fc displayed at 1σ , and red contour is Fo-Fc displayed at 3σ . We state this now in the figure caption.

b) (b) Which active site is it? What type of electronic density map?...

This is active site of subunit B. An omit map is displayed. The caption has been updated accordingly.

c) (d) State the resolution of the map.

The resolution is 2.35 Å.

d) (e)-(g): What is the resolution for the maps?

The resolution is 2.65 Å for soaked crystal data.

The figures have been changed for consistent representation of the maps across the entire text. The caption has been updated appropriately with resolution, contour levels and map types. The resolution information for all the maps is available in the supplementary Table 1.

Note: This figure has been reformatted to supplementary figure 1.

39) Line 542, "Extended data figure 3": (b) what is happening to all the 3 sigma features inside the mask? It is like removing unwanted density...

The map is shown at 2.5 sigma to indicate the level of noise. The pink volume indicates the mask, and the green density for sulbactam is found within the mask. We did not remove any features from inside the mask.

Note: This figure has been reformatted to Figure 7.

40) Line 553, Extended Data Figure 1: S atom colour in the reaction is too light for a white background. Please consider higher contrast.

Figure 1 is updated for better contrast.

REVIEWERS' COMMENTS

Reviewer #2 (Remarks to the Author):

All points raised in the initial review have been addressed.

Reviewer #3 (Remarks to the Author):

The main interrogation I had about the manuscript from Malla et al. was about the physiological form of BlaC. Thanks to the authors answers, BlaC at pH 4.5 is in dimeric form. The question now is to know which dimer is biologically relevant. The authors thinks that the most plausible dimer arrangement is A/B and C/D. In this case, the interactions between residues from chains B/D with the active site of chains A/C are not by-product of crystallographic packing and the kinetic study of the non-covalent intermediate is perfectly biologically relevant. Maybe this point should be emphasized in the manuscript.

In general, I am pleased with the answers the authors gave to the comments. The modifications made by the authors improved the quality of the manuscript. However, I have a couple of minor comments:

- 1) Lines 91-92: It would help if the authors mentioned here the diameter of SUB.
- 2) Lines 121-123: I would add here which dimers are the most probable AB and CD and why.

REVIEWERS' COMMENTS

Reviewer #2 (Remarks to the Author):

All points raised in the initial review have been addressed.

Reviewer #3 (Remarks to the Author):

The main interrogation I had about the manuscript from Malla et al. was about the physiological form of BlaC. Thanks to the authors answers, BlaC at pH 4.5 is in dimeric form. The question now is to know which dimer is biologically relevant. The authors thinks that the most plausible dimer arrangement is A/B and C/D. In this case, the interactions between residues from chains B/D with the active site of chains A/C are not by-product of crystallographic packing and the kinetic study of the non-covalent intermediate is perfectly biologically relevant. Maybe this point should be emphasized in the manuscript.

In general, I am pleased with the answers the authors gave to the comments. The modifications made by the authors improved the quality of the manuscript. However, I have a couple of minor comments:

1) Lines 91-92: It would help if the authors mentioned here the diameter of SUB.

The diameter of SUB (7 Å) is mentioned in line 217.

2) Lines 121-123: I would add here which dimers are the most probable AB and CD and why.

The information has been added to the text in ll. 125 ff.